# Mendelian randomization accounting for complex correlated horizontal pleiotropy while elucidating shared genetic etiology

Qing Cheng [1,2], Xiao Zhang [2], Lin S. Chen [3] ✉ & Jin Liu [2] ✉

Mendelian randomization (MR) harnesses genetic variants as instrumental variables (IVs) to study the causal effect of exposure on outcome using summary statistics from genome-wide association studies. Classic MR assumptions are violated when IVs are associated with unmeasured confounders, i.e., when correlated horizontal pleiotropy (CHP) arises. Such confounders could be a shared gene or inter-connected pathways underlying exposure and outcome. We propose MR-CUE (MR with Correlated horizontal pleiotropy Unraveling shared Etiology and confounding), for estimating causal effect while identifying IVs with CHP and accounting for estimation uncertainty. For those IVs, we map their cis-associated genes and enriched pathways to inform shared genetic etiology underlying exposure and outcome. We apply MR-CUE to study the effects of interleukin 6 on multiple traits/diseases and identify several *S100* genes involved in shared genetic etiology. We assess the effects of multiple exposures on type 2 diabetes across European and East Asian populations.

In the post-genome-wide association study (GWAS) era, many efforts were made to step beyond genetic associations towards causation and mechanistic examinations. Mendelian randomization (MR) assesses the causal effect of potential risk exposures on outcome traits and diseases by leveraging genetic variants as instrument variables (IVs) and integrating existing GWAS summary statistics[1]. MR has been widely applied to study the relationships among complex traits and diseases, and has achieved numerous successes in providing causal evaluations and suggesting disease prevention and therapeutic strategies[2].

Two-sample MR methods take as input two sets of summary statistics, IV-to-exposure and IV-to-outcome association statistics, to estimate the causal effect of exposure on outcome. Since genotypes are 'Mendelian randomized' during meiosis, they are generally not correlated with external unmeasured confounding factors. Classic MR methods imposed strong assumptions on the validity of IVs. They assumed IVs to be associated with the exposure ("relevance"); to affect the outcome only through the exposure ("exclusion restriction"); and

to be unconfounded ("exchangeability"). Figure 1a illustrated the classic assumptions. However, those assumptions are often challenged by the pervasive horizontal pleiotropy − genetic variants affecting outcome via other pathways than exposure. The presence of horizontal pleiotropy can bias the estimation and confound the causal inference if not properly handled. Specifically, the 'uncorrelated horizontal pleiotropy (UHP)' is a phenomenon where a genetic variant affects outcome via other pathways not through exposure (see Fig. 1b left panel for an illustration), and 'correlated horizontal pleiotropy (CHP)' is a phenomenon where a genetic variant affects both exposure and outcome through a heritable shared factor, i.e., an IV being associated with unmeasured confounders (see Fig. 1b right panel). In the recent literature, many robust MR methods were proposed to relax IV assumptions and allow for IVs with UHP either by treating those IVs as outliers[3,4], or by accounting for UHP effects in a model of mixture distributions[5–11]. Some MR methods[12–15] were developed to estimate and adjust for both UHP and CHP. MRMix[12] uses a four-component mixture model to identify and estimate the causal effect using the

[1]Center of Statistical Research, School of Statistics, Southwestern University of Finance and Economics, Chengdu, Sichuan, China. [2]Centre for Quantitative Medicine, Health Services & Systems Research, Duke-NUS Medical School, Singapore, Singapore. [3]Department of Public Health Sciences, The University of Chicago, Chicago, IL, USA. ✉e-mail: lchen@health.bsd.uchicago.edu; jin.liu@duke-nus.edu.sg

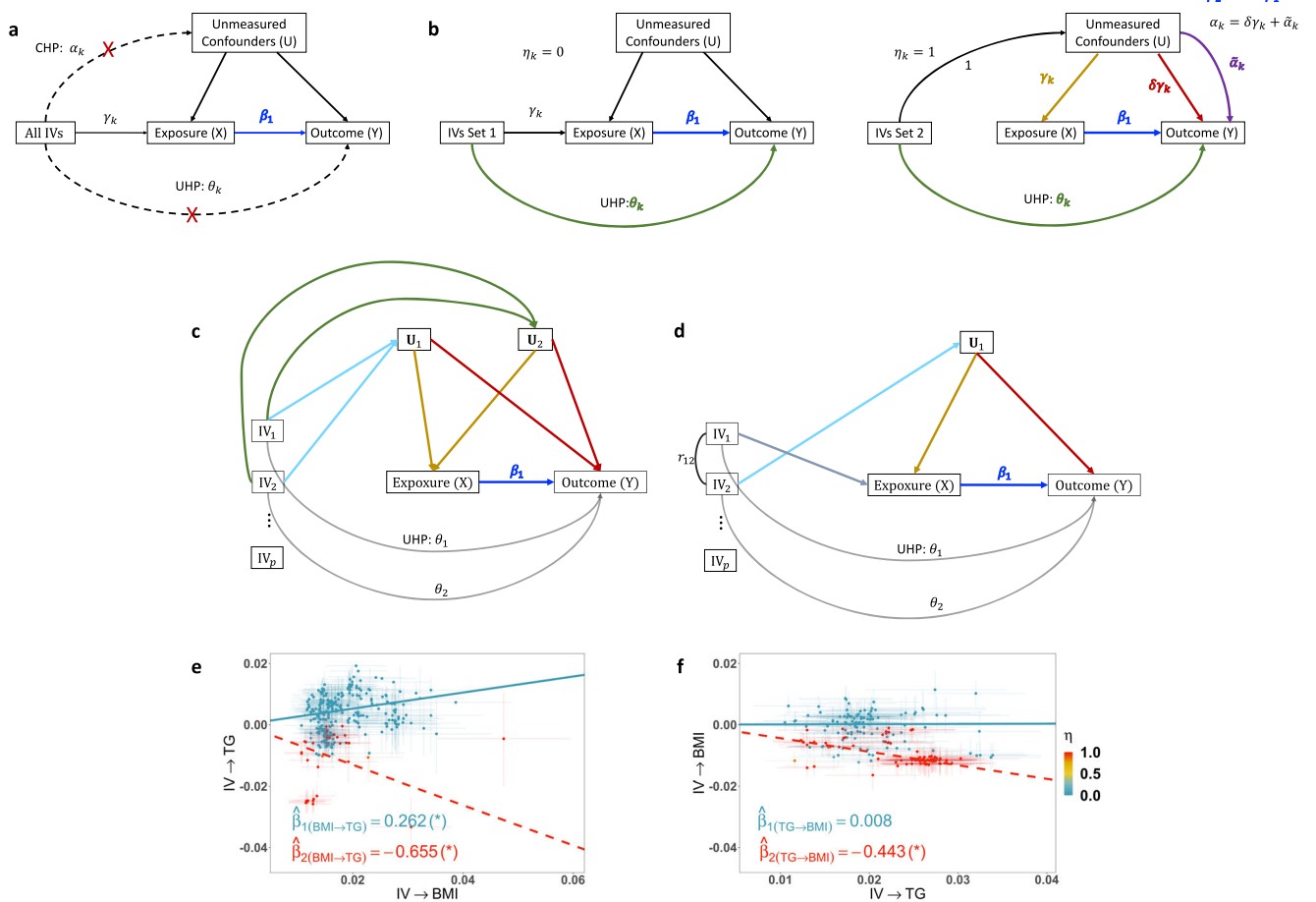

**Fig. 1 | Causal diagrams of classic MR and MR-CUE models, with an illustrative example. a** The causal diagram of classic MR models. Classic MR models assume that IVs affect outcome through only exposure. **b** An illustration of the MR-CUE model. MR-CUE decomposes IVs into two sets, those not affected by CHP (left, $\eta_k = 0$) and those affected by CHP (right, $\eta_k = 1$). MR-CUE allows all IVs to have potential non-zero UHP effect, $\theta_k$. In **b** right panel, we assume that the IV affects the exposure and confounder proportionally, with a sum of IV-to-exposure effect of $\gamma_k$. We rescale the IV-to-confounder effect to be 1 and the effect of confounders on exposure is then $\gamma_k$ (yellow line). The red line represents the decomposed and projected confounder-to-outcome effect and is also proportional to IV-strength, $\gamma_k$. The IV-specific perturbation of confounders may induce an IV-specific bias, $\tilde{\alpha}_k$, which has a mean of zero. **c**, **d** Illustrations of two scenarios when IV-specific CHP effects may arise: **c** there are two or more confounders; **d** there is a single confounder but IVs with different mechanisms are correlated with each other. **e** An illustrative example of estimating the causal effect of BMI on TG in the presence of CHP. **f** The reverse causation estimation of TG on BMI confounded by CHP. When estimating the effect of BMI on TG, some IVs (red) are affected by CHP. Adjusting those IVs would lead to a significant effect estimate of BMI on TG, $\hat{\beta}_{1(BMI \to TG)} = 0.262$, and an insignificant reverse causal effect estimate from TG to BMI, $\hat{\beta}_{1(TG \to BMI)} = 0.008$. In this example, CHP would induce IV-associated confounders and introduce a significant and negative bias. Using estimated IVs with CHP, one may obtain significant causal and reverse causal effect estimates, $\hat{\beta}_{2(BMI \to TG)} = -0.655$, and $\hat{\beta}_{2(BMI \to TG)} = -0.443$.

group of IVs estimated to be valid, without distinguishing the mechanisms (UHP/CHP) of those invalid IVs. CAUSE[13] identifies the IVs with CHP effects, and estimates the causal effect of exposure on outcome using IVs estimated to be not affected by CHP. The method cML-MA[14] uses a constrained maximum likelihood to draw causal inference by excluding IVs with either UHP or CHP. Similar to CAUSE and cML-MA, GRAPPLE[15] assumes the CHP effects (i.e., IV-to-outcome via confounders) being proportional to IV strengths (i.e., IV-to-exposure via confounders). The assumption implies that all IVs perturb the whole confounder set and further affect outcome under a same mechanism, differing by only IV strengths.

Correlated horizontal pleiotropy is a challenging and frequently occurring issue in MR analyses. When there is only one confounder, all IVs with CHP affect the same confounder and the CHP effects of different IVs are proportional to IV strengths. Existing methods[13–15] consider and model the shared CHP effect for all IVs. Often for complex traits and diseases, many genes and pathways (e.g., metabolism, immune pathways) may affect both exposure and outcome. In this work, we propose a MR method, MR-CUE (MR with Correlated

horizontal pleiotropy Unraveling shared Etiology and confounding). MR-CUE accounts for more complex and realistic CHP effects in the presence of multiple confounders and by leveraging correlated IVs to boost power. As illustrated in Fig. 1b right panel, for IVs affected by CHP, we set the effect of IV-to-confounder to be 1[13], confounder-to-exposure to be $\gamma_k$, and confounder-to-outcome effect to be $\alpha_k$. When estimating the causal effect from exposure on outcome, CHP induces a bias and the bias is equal to the shared CHP effect parameter on outcome, $\delta = E(\frac{\alpha_k}{\gamma_k})$. If unbalanced CHP is present ($\delta \neq 0$) and unadjusted, false positives may arise or power may be reduced. We propose that the effect of confounder set on outcome can be decomposed into two parts, $\alpha_k = \delta\gamma_k + \tilde{\alpha}_k$. The first part is the shared confounding effect across all IVs with CHP and is proportional to the confounders' effect on exposure ($\gamma_k$) induced by each IV; and the second part ($\tilde{\alpha}_k$) captures how IV-specific perturbation to confounder set may affect outcome, and is orthogonal to the first part. When there exist multiple confounders (Fig. 1c), different IVs may be associated with multiple confounders at different strengths, and those IVs perturb the confounder set differently. For each IV, the ratio $\alpha_k/\gamma_k$ is a weighted average among

all confounders, and the ratios are not a constant for all IVs. Additionally, the inclusion of correlated IVs in MR analyses increases the number of instruments and may boost the power[8]. When there are multiple correlated IVs and even if there is only one confounder (Fig. 1d), the correlations among IVs with different mechanisms may induce IV-specific CHP effects. The issue is insufficiently addressed in the existing literature when using correlated IVs. Figure 1e, f shows a real data example in which CHP is present between body mass index (BMI) and triglycerides (TG). Without properly identifying and accounting for complicated CHP effects, the effect of exposure on outcome could be confounded and the estimated causal effect of outcome on exposure is also non-zero, i.e., reverse causation may occur. By modeling both the shared CHP and IV-specific CHP effects, MR-CUE estimates the causal effect and distinguishes it from reverse causation. Moreover, the modeling of IV-specific CHP effects alleviates the potential bias in the presence of many weak instruments.

Another feature of MR-CUE is that we propose to further study sets of IVs estimated to have CHP and examine their cis-associated genes and involved pathways. In contrast to existing method[15], MR-CUE allows for overlapping genes/pathways. It provides the quantification of estimation uncertainty in identifying IVs with CHP and allows us to further study the sets of IVs estimated to have CHP at different levels of confidence. Through two examples, we illustrate that the estimated IVs/variants with CHP can suggest genes and pathways that are suspected sources of IV-associated confounders. Those genes and pathways may shed light on the shared genetic etiology for traits and diseases affected by a common exposure, or may reveal relevant pathways and mechanisms underlying different causal exposures for a complex disease outcome. Those disease-relevant common confounders and pathways could inform concerted mechanisms and etiologies across populations and ethnic groups.

## Results

### MR-CUE examines causal effects by delineating correlated and uncorrelated horizontal pleiotropic effects

We propose MR-CUE to estimate the causal effect from exposure ($X$) on outcome ($Y$) while accounting for both UHP and CHP. As illustrated in Fig. 1b, we model the IV-to-outcome effect of the $k$-th IV ($k = 1, ..., p$), $\Gamma_k$, as a function of IV-to-exposure effect, $\gamma_k$, and pleiotropic effects:

$$\Gamma_k = \begin{cases} \beta_1 \gamma_k + \theta_k, & \text{if } k \in \text{IV Set 1 with no CHP} \\ \beta_1 \gamma_k + \theta_k + \alpha_k, & \text{if } k \in \text{IV Set 2 with CHP,} \end{cases} \quad (1)$$

where $\beta_1$ is the causal effect of exposure on outcome; $\theta_k$ is the UHP effect, and $\alpha_k$ is the CHP effect of the $k$-th IV; and both the IV-to-outcome and IV-to-exposure effects, $\Gamma_k$ and $\gamma_k$, respectively, can be obtained from GWASs. We assume that all IVs may have UHP effects, $\theta_k$, while only a proportion of IVs may also have CHP effects. Following existing literature[13], we rescale the IV-to-confounder effect to be 1 and the effect of confounders on exposure is then $\gamma_k$. In Fig. 1b (right panel), the line representing the direct effect from IV to exposure is omitted to avoid over-parameterization since it is assumed to change proportionally with IV-to-confounder effect. As discussed before, we decompose the CHP effect into two components, $\alpha_k = \delta \gamma_k + \widetilde{\alpha}_k$, representing IV-shared and IV-specific CHP effects. We reparametraize our model as

$$\Gamma_k = \begin{cases} \beta_1 \gamma_k + \theta_k, & \text{if } k \in \text{IV Set 1 with no CHP} \\ \beta_2 \gamma_k + \theta_k + \widetilde{\alpha}_k, & \text{if } k \in \text{IV Set 2 with CHP,} \end{cases} \quad (2)$$

where $\beta_2 = \beta_1 + \delta$ is a nuisance parameter capturing both $\beta_1$ and $\delta$, and $\delta$ is the IV-shared confounding parameter due to CHP. For IVs in Set 2, the IV-specific CHP effect, $\widetilde{\alpha}_k$, is assumed to have a Gaussian prior. By accounting for IV-specific CHP effects (i.e., IV-specific perturbations to the confounder set), our model is robust to the presence of multiple

confounders without explicitly modeling the effect of each confounder. MR-CUE is built on a Bayesian hierarchical model that estimates the parameters from the above model and obtains inference via Gibbs sampling. In Fig. 1e, we illustrate our model using a real data example to assess the causal effect of BMI on TG. When plotting IV-to-BMI effects against IV-to-TG effects in Fig. 1e, there is a positive causal relationship for some IVs (blue) while there are a few other IVs entailing a different pattern with an opposite slope (red). The proposed MR-CUE model identifies the IVs affected by CHP (red dots), and estimates the causal effect from BMI on TG using IVs not affected by CHP (blue dots). The unconfounded causal effect is estimated to be significant and positive, $\hat{\beta}_{1(\text{BMI}\rightarrow\text{TG})} = 0.262$. For IVs affected by CHP, their estimated causal effects is significant and negative, $\hat{\beta}_{2(\text{BMI}\rightarrow\text{TG})} = -0.655$, due to the large and negative confounding bias $\delta$. As further illustrated in Fig. 1f, MR-CUE reduces false positive findings due to reverse causation by identifying the IVs affected by CHP and quantifying the uncertainty in the estimation/identification. Without properly handling CHP, one may obtain a crude sum of effect estimates combining the unconfounded and the confounded effects. In the BMI-TG example, we observe that the combined effects (red), $\hat{\beta}_2$'s, for both BMI-to-TG and TG-to-BMI are significant and negative, due to the shared confounding. While the unconfounded effect is only significant from BMI to TG, not the reverse. In the presence of unadjusted CHP, one may suffer from a reduced power or an inflated type I error rate depending on the direction of confounding effect.

In practice, there is often no clear cut for IVs unaffected or affected by CHP due to trait polygenicity and LD. The uncertainty of each variant belonging to either IV Set 1 or Set 2 can be accounted for by modeling a latent variable, $\eta_k$. MR-CUE imposes a spike-slab prior[16,17] for $\widetilde{\alpha}_k$, with a spike (mass density) at zero and a slab spreading over a wide range of plausible values. MR-CUE quantifies the probability of each variant being affected by CHP. Different than existing clustering-based methods or methods involving the selection of IVs estimated to be valid, MR-CUE provides the estimated probabilities of IVs from Set 1 or Set 2. MR-CUE obtains the causal effect estimate as a weighted estimator from all IVs weighing by the posterior probabilities of IVs being from Set 1. With the estimated probabilities of IVs from Set 2, MR-CUE also works as a useful tool for further examining the potential shared genetic components underlying exposure and outcome. The IVs estimated to have CHP and their cis-associated genes may imply common genes and genetic pathways associated with both exposure and outcome. To further allow IVs in LD, MR-CUE partitions the whole genome into independent blocks and introduce a group latent variable, $\eta_l$, for IVs in same blocks (see Methods).

### MR-CUE identifies IVs with CHP effects, estimates the causal effects and reduces false positives

We conducted simulation studies to evaluate the performance of MR-CUE and compare with existing MR methods in a variety of scenarios. We first generated genotype matrices from different LD patterns (Methods section). Both exposure and outcome were simulated based on polygenic architecture as shown in Eq. (13). In simulations, we considered both single and multiple confounders (Methods section and Supplementary Materials). All IVs ($p = 1000$ or $2000$) contributed a total heritability of 0.1 to exposure, while the heritability for outcome can be decomposed as variation through the causal effect ($\beta_1$), variation contributed by UHP ($\theta$), and variation attributable to CHP ($\alpha$). We controlled the combinatorial values for heritability due to UHP and CHP, denoted as $h_\theta^2$ and $h_\alpha^2$, respectively. As discussed earlier, we assumed that CHP is due to shared genetic components between exposure and outcome traits and only a proportion of IVs have non-zero CHP effects. We performed single-variant association tests to obtain the summary statistics for both IV-to-exposure and IV-to-outcome associations as input for MR analyses.

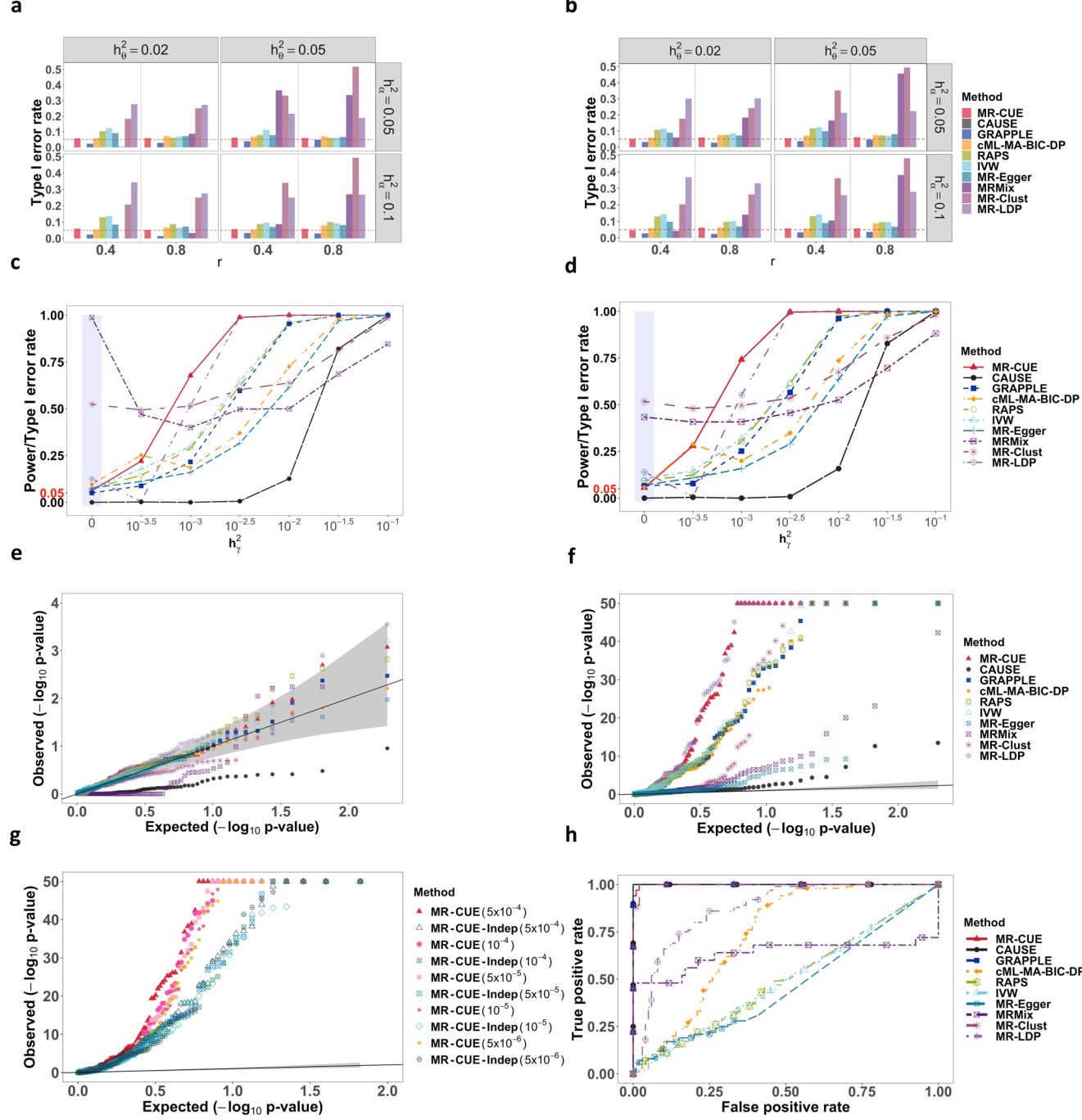

**Fig. 2 | Comparison of MR-CUE and other MR methods in simulation studies.**
**a**, **b** Type I error rates for MR-CUE and other methods under combinatorial settings for $h_\theta^2$ and $h_\alpha^2$ with $\rho_{\alpha\gamma} = 0.2$ and $p = 1000$ for single and multiple confounders, respectively. **c**, **d** Powers for MR-CUE and other methods under the setting: $h_\theta^2 = 0.1$, $h_\alpha^2 = 0.05$, $p = 1000$, $r = 0.4$ and $\rho_{\alpha\gamma} = 0.2$ for single and multiple confounders, respectively. **e** QQ plots of $-\log_{10}(p\text{-values})$ for all methods under the null from analyses of negative controls. **f** QQ plots of $-\log_{10}(p\text{-values})$ for all methods from analyses of positive controls. The $p$-values of all methods are two-sided without multiple testing adjustment. **g** QQ plots of $-\log_{10}(p\text{-values})$ for MR-CUE with correlated and independent IVs. The gray regions in **e**–**g** indicate 95% confidence intervals. **h** ROC curves for evaluation of causation and reverse causality among all methods.

We compared MR-CUE with nine other methods, including CAUSE[13], GRAPPLE[15], cML-MA[14], RAPS[6], IVW[18], MR-Egger[5], MRMix[12], MR-Clust[10], MR-LDP[8]. In existing literature, other methods including BESIDE-MR[19], JAM-MR[20], Berzuini's method[21], and MR-Corr2[22] have also been proposed to account for either UHP or CHP. For cML-MA, we evaluated its performance using its default setting, cML-MA-BIC-DP. Among those methods, MR-LDP, RAPS, IVW, MR-Egger, and MR-Clust assumed that no IV/variant is affected by CHP, but allowed IVs to have UHP effects. The proposed MR-CUE and four other methods, i.e.,

CAUSE, GRAPPLE, cML-MA, and MRMix, allowed IVs to have both UHP and CHP. Among all competing methods, MR-CUE and MR-LDP can handle variants in moderate-to-strong LD, and CAUSE allowed for variants in weak LD.

First, we evaluated the performance of type I error rate control (Fig. 2a, b) for all competing methods in the scenarios of both single and multiple confounders. In both scenarios, MR-CUE could sharply control type I error rates in all settings while CAUSE, GRAPPLE and cML-MA-BIC-DP had a reasonable control of the type I error rates.

CAUSE and GRAPPLE were conservative in many settings while cML-MA-BIC-DP could control the type I error rate at the expenses of power reduction (Fig. 2c, d). Since MR-Clust and MR-LDP did not account for CHP effects, their type I error rates were inflated. We also observed inflated type I error rates for MRMix. Simulations for all methods except for MR-CUE and MR-LDP were based on independent IVs after SNP clumping, since those methods were initially proposed using IVs in weak-to-moderate LD. With independent IVs, RAPS, IVW and MR-Egger could generally control the type I error rates, up to some slight inflation. We also performed simulations with a larger number of IVs ($p = 2000$) and a stronger correlation between IV-to-exposure and CHP effects, $\rho_{\alpha\gamma}$. The results were largely similar, and additional details were provided in Supplementary Fig. 1. When the correlation in CHP ($\rho_{\alpha\gamma}$) was stronger, RAPS, IVW and MR-Egger suffered from increased levels of inflation in the type I error rates. Supplementary Figure 2 compares the estimation biases of MR-CUE with other methods and shows the boxplots of point estimates for competing methods.

We compared the power of each method by varying $h_{\gamma}^2$ while fixing $h_{\theta}^2 = 0.1$, $h_{\alpha}^2 = 0.05$, $r = 0.4$, and $\rho_{\alpha\gamma} = 0.2$, with single or multiple confounders (Fig. 2c, d). MR-CUE achieved the highest power among the methods that could control the type I error rates. CAUSE, as a conservative method, was under-powered[13] and cML-MA-BIC-DP was less powerful than MR-CUE. We also considered other simulation settings with different $h_{\theta}^2$, $h_{\alpha}^2$, autoregressive coefficient $r$ for LD, and correlation $\rho_{\alpha\gamma}$ in CHP. Results were similar, and additional details were provided in Supplementary Figs. 3–6.

Next, we evaluated the performance of MR-CUE in selection/identification of IVs with CHP effects. MR-CUE provided a quantitative metric for this purpose. We considered two prior distributions, i.e., the default prior (a Beta distribution with shape parameters being 2 and $L$, the number of LD blocks) and the non-informative prior, Beta(1,1). Here, we considered $h_{\theta}^2 = 0.02$ or $0.05$, $h_{\alpha}^2 = 0.05$ or $0.1$, the correlation between $\alpha_k$ and $\gamma_k$ being $\rho_{\alpha\gamma} = 0.2$ or $0.8$, and causal effect $\beta_1 = 0$ or $0.1$ with $p = 1000$ or $2000$. Note that when $\rho_{\alpha\gamma} = 0$, only UHP is present. We also considered moderate and strong LD structure ($r = 0.4$, $0.8$) with autoregressive correlation. Supplementary Figure 7 shows the false discovery rate (FDR) for identifying IVs with CHP effects and Supplementary Fig. 8 shows the corresponding area under the curve (AUC) of the receiver operating characteristic (ROC) curve. MR-CUE with the default prior can control the FDR at the nominal level of 0.1 while achieving a high level of AUC.

We evaluated the performance of MR-CUE and other methods using real data with negative and positive controls[23], with varying IV selection thresholds. In the analyses of negative control outcomes, we used self-reported tanning ability and hair color as outcome, since both traits were largely determined at birth and were unlikely to be affected by other traits we considered[24]. We considered 16 complex traits and diseases (Supplementary Data 1a) as exposure to evaluate the control of type I error rates for MR-CUE and other MR methods. For each method, we applied five different IV selection thresholds to evaluate the sensitivity of different methods to IV selection criteria. Figure 2e shows the quantile-quantile (QQ) plot of negative log base 10 of $p$-values for MR-CUE and other methods when IV selection threshold was $5 \times 10^{-4}$. MR-CUE and some existing MR methods including GRAPPLE, cML-MA-BIC-DP, RAPS, IVW and MR-Egger can well control type I error rates, with $p$-values falling within the 95% confidence band of the null distribution. Note that in the analyses of negative control outcomes, some MR methods without considering CHP performed well. This was probably because that the outcomes considered were not polygenic and there was no CHP effects. On the other hand, MR-LDP had slightly inflated $p$-values while CAUSE, MRMix, and MR-Clust had deflated $p$-values. In the analyses of positive controls, we selected 100 established pairs of traits and diseases with causal relationships supported by exiting literature. The pairs of exposure and outcome were listed in Supplementary Data 1b. We also applied different IV

selection thresholds to evaluate the sensitivity of results to IV selection. Figure 2f shows the QQ plots of negative log base 10 of $p$-values using $5 \times 10^{-4}$ as the IV selection threshold. The QQ plots using other thresholds and only independent IVs were provided in the Supplementary Figs. 13–15. In all scenarios, MR-CUE had the highest power. MR-LDP also had high powers but suffered from inflated type I error rates as shown in both simulations and negative control analyses. Figure 2g shows the QQ plots of positive control for MR-CUE using correlated and independent IVs, respectively. We observed a substantial power gain of the proposed MR-CUE with correlated IVs and with relaxed IV selection thresholds. Last, we evaluated whether MR-CUE could distinguish causal relationship from reverse causality. Reverse causality occurs when there exist IVs affecting the exposure and outcome traits through some shared confounding factors. Since MR-CUE is capable of identifying IVs with CHP effects, it is expected to identify the direction of true causal effect and reduce false positive findings due to reverse causality. To examine this, we simulated data with a causal effect from a trait A on a trait B ($\beta_{A \to B} \neq 0$), and tested for a reverse causal effect from B on A (B → A) using MR-CUE and other methods. The simulation details were provided in the Methods Section. In all scenarios, we fixed the heritability for exposure and outcome at 0.3 and 0.25, respectively. For each simulation replicate, we applied the above MR methods for assessing the causal effects in both directions. We evaluated and compared the powers for detecting the true causal effect of exposure A on outcome B, while also compared the type I error rates for the reverse causal effect of outcome B on exposure A. Figure 2h shows the ROC curves using 100 simulated replicates at varying significance thresholds. MR-CUE, CAUSE, GRAPPLE, and MR-Clust could distinguish causal effects from reverse causation in all simulations, while other methods cannot.

Results from other considerations, including non-linear confounding effects, binary outcome, the impact of different proportions in IVs with CHP effects, and a sparse vector for UHP in reverse causation, showed similar conclusions and can be found in Supplementary Figs. 9–12.

## Examining the effects of interleukin 6 on multiple traits/diseases implies shared genes and pathways as sources of CHP

Interleukin 6 (IL-6) is a key inflammatory cytokine, and has both pro- and anti-inflammatory properties. It plays an important role in immune-related processes and pathways[25]. Here we applied MR-CUE and other MR methods to evaluate the causal effects of IL-6 on 27 complex traits and diseases (Supplementary Data 1c). The soluble IL-6 receptor (sIL6R), a negative regulator of IL-6 signaling, has been suggested to affect many complex traits and diseases including lipid levels (e.g., high-density lipoprotein cholesterol, HDL-c), both severity and susceptibility of COVID-19, heart diseases (e.g., atrial fibrillation, AF), autoimmune diseases (e.g., Crohn's disease, CD), and others[25,26]. We analyzed those complex traits/diseases and other diseases that may not be affected by IL-6. Supplementary Table 3 and Supplementary Data 2a summarize the $p$-values and the estimated causal effects for MR-CUE and other methods.

IL-6 is a multifunctional cytokine and is highly polygenic with a heritability estimate of up to 61%[27]. In addition to estimating the causal effects of IL-6, we further obtained the posterior probabilities of IVs having CHP effects on each of the 27 outcomes, $\Pr(\eta_l = 1 | \text{data})$, from each chromosome clustered in blocks. In Fig. 3a right panel, we plotted the strengths of CHP effects for IVs across all chromosomes for 27 outcomes, with estimated causal effects shown in the very right column. In Fig. 3a left panel, we also plotted the genetic correlations among 27 outcome traits estimated by LDSC[28]. From the heatmap, we observed that traits in high genetic correlations tend to have similar or dependent estimated causal effects of IL-6, e.g., COVID19 severity and susceptibility; any stroke (AS), any ischemic stroke (AIS), and cardioembolic stroke (CES). Those outcomes also presented similar

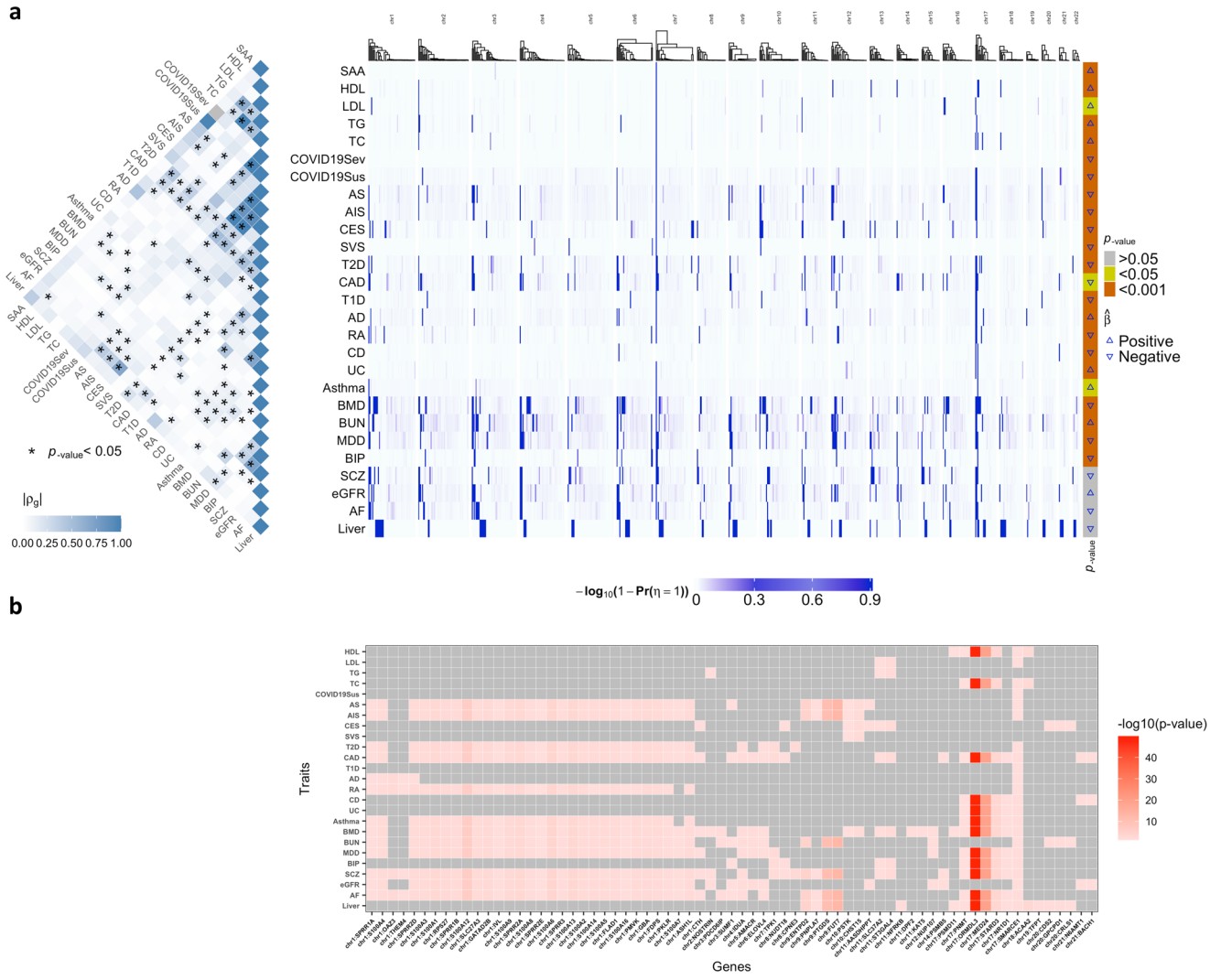

**Fig. 3 | MR-CUE analysis of IL-6 on multiple traits/diseases. a** (Left panel) The heatmap of the estimated genetic correlations ($\rho_g$) among the 27 examined outcomes with IL-6 as exposure. The genetic correlation $p$-values are from two-sided LDSC tests[28] without multiple testing adjustment. (Right panel) The heatmap of the estimated strengths of CHP, $-\log_{10}(1 - \mathrm{Pr}(\eta_l = 1))$, for selected IVs across all chromosomes for the 27 outcomes. The $p$-values on the right bar indicate the significance of the causal effects of IL-6 on the examined outcomes, and are from two-sided MR-CUE tests without multiple testing adjustment. **b** The heatmap of a partial list of cis-genes that were significantly associated with at least one IV affected by CHP across multiple outcomes, with color indicating the strength of the most significant association for each gene. Cis-associations were assessed using blood tissue samples from the Genotype-Tissue Expression (GTEx) project for IVs with estimated CHP effect, with nominal $p$-values from two-sided Pearson correlation tests.

patterns of CHP effects. Note that the strong correlation between COVID19 susceptibility and severity may be artificial due to selection bias, since people with more severe COVID19 infection are also more likely to be diagnosed with COVID19. On the other hand, traits in mild-to-moderate genetic correlations, e.g., bone mass density (BMD), blood urea nitrogen (BUN), major depressive disorder (MDD), bipolar disorder (BIP), and schizophrenia (SCZ), may not share causal effect estimates but could still share CHP effect patterns. CHP effects could be present when there are no causal effects.

We further identified the IVs with significant CHP effects, $\mathrm{Pr}(\eta_l = 1|\mathrm{data}) > 0.8$, and examined the genes in cis (1MB distance) and being associated with those IVs ($p$-value < 0.05). The identified genes and gene sets may shed light on the shared pathways between IL-6 and the examined complex outcomes. In Fig. 3b, we plotted the heatmap of selected cis-genes associated with at least one IV affected by CHP across multiple outcomes, with color indicating the strength of the most significant association of the gene and its cis-IVs with CHP. There were many genes involved in the same pathways and being identified as IV-associated shared factors across multiple outcomes. Those

shared genes may partially explain the observed genetic correlations among those 27 traits/diseases in Fig. 3a (left panel). Specifically, MR-CUE identified 13 *S100* genes encoding S100 proteins located in the chromosome 1q21 region. The S100 proteins belong to a family of calcium-binding cytosolic proteins and have a broad range of intra-cellular and extracellular functions. The extracellular S100 proteins play a crucial role in the regulation of immune homeostasis, post-traumatic injury, and inflammation[29]. S100 proteins trigger inflammation through their interactions with receptors for *RAGE* and *TLR4*[30]. *S100A12* has been shown to induce the production of pro-inflammatory cytokine IL-6 and IL-8 in both a dose-dependent and time-dependent manner[29]. Additionally, S100 proteins play a significant role to the development of chronic inflammatory and auto-inflammatory diseases[31,32]. MR-CUE also identified some genes in cor-nified envelope pathway, *SPRR* family and *IVL*. These genes together with *S100* genes constituted the epidermal differentiation complex that are essential for epidermal differentiation, building the first-line defense against external assaults and protecting our bodies from dehydration[33]. Genes in ATPase complex were identified to play a

shared role as well. Existing literature[34] reported that the over-expression of *KAT5* gene potentiated transcription of downstream antiviral genes including *IL-6*. Other works[35] reported that histone methyltransferase *ASH1L* suppresses *TLR*-induced *IL-6* production.

The above analysis also showed that different IVs with CHP effects may be involved in multiple pathways entailed by multiple sources of IV-associated confounders. The confounding effect on outcome could be IV-specific. MR-CUE allows the estimation of an overall CHP effect while accounting for IV-specific variation/perturbation to confounders and improves the estimation of CHP. By closely examining the IVs with CHP effects and their cis-associated genes, we identified genes and gene sets that were highly inter-connected as suggested sources of IV-associated confounders and further informed potential shared genetic etiology among the traits examined.

## MR-CUE informs type 2 diabetes-related pathways for multiple risk factors across two populations

We applied MR-CUE to each exposure-T2D trait pair and separately estimated the causal effect from each exposure on T2D risk in the European and East Asian populations. Type 2 diabetes (T2D) is a form of diabetes characterized by high blood sugar, insulin resistance, and relative lack of insulin[36]. T2D is high polygenic and has a complex etiology[37,38]. Examining multiple potential risk exposures for T2D may reveal common patterns in the etiology for related factors while also presenting unique characteristics for different types of factors. Established risk factors for T2D include both lifestyle factors, such as overweight and obesity, and medical conditions[39]. We also considered other exposure traits, including lipid levels, e.g., TG and high-density lipoprotein cholesterol (HDL-c), blood cell parameters, e.g., counts for red blood cells (RBC) and white blood cells (WBC), insulin-resistance-related factors, e.g., fasting insulin (FI), fasting glucose (FG) and HbA1c, and others. We examined 29 and 14 exposures for T2D in European and East Asian populations, respectively. The full list of exposure traits/diseases was provided in the Supplementary Data 1d, e. Supplementary Tables 4 and 5 and Supplementary Data 2b, c summarize the *p*-values and the estimated causal effects for MR-CUE and other methods. We further pulled the results from MR-CUE and the estimated sets of IVs with CHP across analyses of different exposures to examine shared confoundings and mechanisms in both populations. Some exposures for T2D are significant in both populations, such as obesity and blood cell parameters. Obesity is a well-known risk factor for T2D and the associations of blood cell parameters and T2D were also reported in many studies[40–42]. HbA1c was also identified by MR-CUE in both populations and its association with hypoglycemia was reported in a previous study[43]. Some established T2D risk factors, including insulin resistance, insulin-resistance-related factors, and other obesity factors, have genetic-association summary statistics in only the European population, and thus the cross-population comparison was not presented. MR-CUE reported significant causal effects for those factors in the European population. Cross-populations analyses using summary statistics from different populations and ethnic groups still present many challenges due to the substantially varying LD patterns, difficulties in data harmonization, study heterogeneity and others. Moreover, only a proportion of the causal variants and genes for complex traits/diseases might be shared across populations, and the risk exposures for a complex disease could also differ by population. MR-CUE is robust in cross-population analyses as it offers two layers of inference – it obtains the causal effect estimation using IVs not affected by IV-associated confounders, while also maps the underlying genes and pathways for IVs affected by confounding.

To further investigate the shared genetic pathways for the 29 and 14 traits in the European and East Asian populations, we obtained the IVs with significant CHP effects, $\Pr(\eta_l = 1|\text{data}) > 0.8$. In Fig. 4a, b, we plotted the strengths of CHP effects for IVs across chromosomes in both European and East Asian populations, respectively.

In general, exposures with higher polygenicity tend to have more IVs with CHP. We further performed pathway analysis based on those IVs using SNPnexus[44] and obtained their enriched pathways, shown in Fig. 4c, d for European and East Asian populations, respectively. The significant causal risk factors identified by MR-CUE are similar in both populations, and the enriched pathways presented some cross-population similarity as well. MR-CUE identified both metabolism and immune response pathways for multiple exposures and T2D in both populations. T2D itself is an inflammatory disease triggered by disordered metabolism[45]. MR-CUE identified many metabolic-related factors, including glycine, fasting glucose, and fasting insulin, having shared genetic components in metabolism pathway with T2D. Dys-regulation of lipid metabolism triggers *NLRP3* activation leading to obesity-induced inflammation and insulin resistance[46,47]. Moreover, HbA1c that is chemically linked to a sugar was used as a screening tool to detect early T2D[48]. Fasting glucose and HbA1c shared many common pathways in European population (Fig. 4c) while pathways for HbA1c were similar in both populations. A recent work[49] reported that genetic variants in glutamate cysteine ligase conferred protection against T2D, while glycine was considered a promising amino acid for improving metabolic health[50]. Glutamate and glycine are both metabolites, and they play critical roles in the metabolism pathway. Glycine was reported to improve immunity and treat metabolic disorders in diabetes[51], while glutamate was found to be a key immunomodulator in the initiation and development of T-cell-mediated immunity[52]. We also observed that many exposures share the signal transduction pathway with T2D in both populations. Signal transduction pathway plays an important role in both red blood cell[53] and T2D[54,55]. Biologically, signal transduction contains insulin receptor signaling pathway that may mediate the development of T2D by endoplasmic reticulum stress[56]. MR-CUE assessed the causal effect of each risk exposure on T2D risk, while other T2D-related exposures are potential confounders and may contribute to the CHP effect. An alternative and complementary analysis may be using a multivariable MR method to jointly examine the effect of multiple exposures. Most existing multivariable MR methods assume no CHP, i.e., all IV-associated confounders being accounted for, and we did not proceed this direction.

## Discussion

In this work, we propose MR-CUE to obtain causal inference accounting for both UHP and CHP in complex and realistic settings. When there are multiple confounding genes affecting both exposure and outcome, different IVs may be associated with more than one confounder at varying levels of strengths, resulting in both IV-shared and IV-specific CHP effects. In contrast to existing methods focusing on IV-shared CHP effects, MR-CUE also models IV-specific CHP effects, and estimates the causal effect of exposure on outcome. Moreover, MR-CUE allows moderately correlated IVs to boost power in MR analyses. When correlated IVs are included, IV-specific CHP effects may also arise. Existing methods insufficiently address the issue, while MR-CUE can obtain unbiased and efficient estimation in the presence of multiple confounders and/or correlated IVs. MR-CUE simultaneously quantifies the probabilities of IVs with CHP, and further examines their cis-associated genes for potential shared genes/pathways/mechanisms underlying exposure and outcome. With simulation studies and analyses of negative control outcomes and positive controls, we demonstrated that MR-CUE can reduce false positives due to reverse causation, control the type I error rates in the presence of multiple confounders and correlated IVs; by including correlated IVs, MR-CUE improves the power of MR analyses; MR-CUE is insensitive to IV selection threshold; and MR-CUE identifies IVs with CHP at the desired confidence levels. To minimize potential bias due to the winner's curse, we recommend selecting the IVs first using a third independent sample[57], if possible.

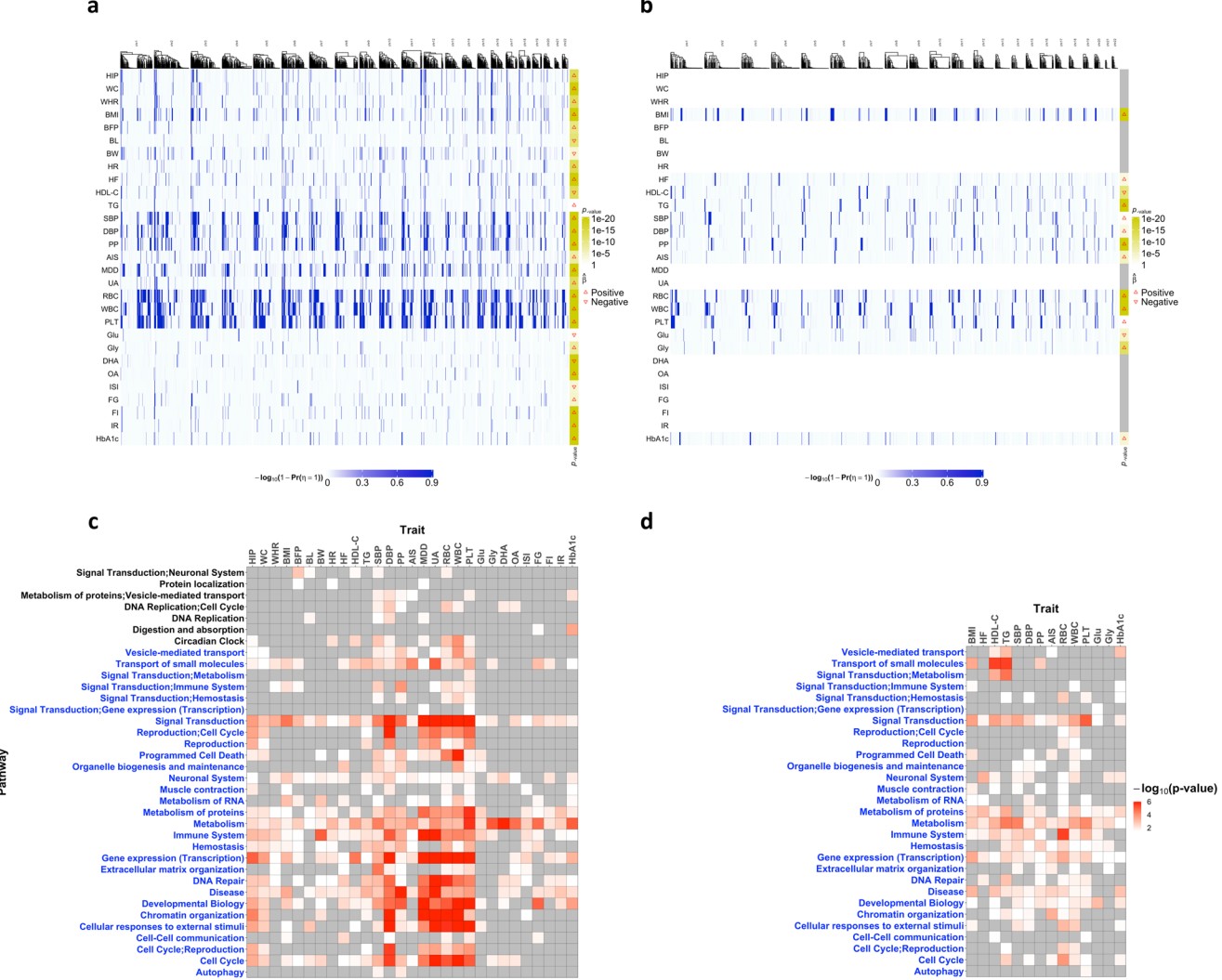

**Fig. 4 | MR-CUE analysis of exposure-T2D trait pairs. a, b** The heatmaps of the estimated strengths of CHP, $-\log_{10}(1 - \Pr(\eta_l = 1))$, for selected IVs across all chromosomes for 29 and 14 exposures for T2D in the European and East Asian populations, respectively. The *p*-values are calculated based on two-sided MR-CUE tests without multiple testing adjustment. **c, d** The heatmaps of enriched pathways for identified IVs with CHP by exposure in the European and East Asian populations, respectively. The *p*-values are calculated based on one-sided Fisher's exact tests without multiple testing adjustment. The blue y-axis in **c, d** represent the common pathways of European and East Asian populations.

We studied the causal effects of IL-6 on multiple outcomes. By further examining the IVs with significant CHP effects and their cis-associated genes, we highlighted multiple genes that may be shared (also served as confounders) between IL-6 and some examined traits/diseases. Those suggested genes included multiple *S100* genes and genes in the cornified envelope pathway, shedding light on the shared genetic etiology. In another analysis, we applied MR-CUE to study the effects of multiple putative exposures on T2D risks in both European and East Asian populations. A cross-population analysis and comparison of multiple risk exposures showed consistent causal effect estimates in both populations. We further examined the IVs with CHP effects and their enriched pathways. In both populations, it was suggested that metabolism and immune response pathways play a central role in the shared etiologies among multiple putative exposures and T2D.

MR-CUE paved the way for future cross-population MR analyses to reduce disparity. Cross-populations MR analyses using summary statistics from different populations is still challenging due to varying LD patterns, difficulties in data harmonization, study heterogeneity and others. MR-CUE is robust in cross-population analyses as it provides double layers of inference for cross-population comparisons – it estimates the causal effect of exposure using IVs not affected by IV-associated confounders, while also maps the underlying genes and pathways for IVs affected by confounding.

MR-CUE has some caveats that may require further explorations. First, MR-CUE assumes that all IVs could have potential UHP effect while only a sparse proportion of IVs have CHP effect. When the proportion is non-sparse, the identification condition may lead to biased estimation. Second, MR-CUE works for a single exposure and a single outcome. When the exposure is known to be highly correlated with other exposures, or when multiple outcomes may often co-occur, multi-variable MR methods accounting for both CHP and UHP may be considered. Third, MR-CUE requires multiple (at least dozens of) IVs to identify and delineate CHP effects and is not suitable for analyzing molecular risk exposures such as gene expression levels. Last, MR-CUE identifies the IVs with significant CHP effects, though the mapping of cis-associated genes/pathways from those identified IVs is still not an automated process. We are working on improving the automation of this step.

When using MR to infer causation, caution should always be exercised. By leveraging GWAS summary statistics from large genetic consortia or biobank-sized studies, MR analysis is empowered. On the

other hand, insights are still limited regarding potential subgroup effects, indirect effects from different mediators between exposure and outcome, and potential exposure-mediator interactions. Further integration of MR with mediation analyses could be valuable for the development of prevention and treatment strategies towards precision medicine.

## Methods

### MR-CUE model for independent IVs

To estimate the causal effect in the MR-CUE model, we use the marginal effect size and standard error estimates from GWASs for exposure ($X$) and outcome ($Y$) diseases/traits as input. Let $\{\widehat{\gamma}_k, \widehat{\mathbf{s}}_{\gamma_k}\}$ denote the IV-to-exposure effect size and its standard error for IV $k$. Let $\{\widehat{\Gamma}_k, \widehat{\mathbf{s}}_{\Gamma_k}\}$ denote the IV-to-outcome effect size and standard error. Let $\gamma_k$ and $\Gamma_k$ be the true marginal effect size of IV $k$ for traits $X$ and $Y$, respectively. For independent IVs, we model the distribution for the estimated effect sizes in both exposure and outcome diseases/traits using the following independently and identically distributed (*i.i.d.*) normal distributions,

$$\widehat{\gamma}_k \sim \mathcal{N}(\gamma_k, \widehat{\mathbf{s}}_{\gamma_k}^2), \text{ and } \widehat{\Gamma}_k \sim \mathcal{N}(\Gamma_k, \widehat{\mathbf{s}}_{\Gamma_k}^2). \tag{3}$$

The proposed MR-CUE models the IV-to-outcome effect as a function of IV-to-exposure, and UHP and CHP effects using Eq. (2), with UHP effects *i.i.d.* as $\theta_k \sim \mathcal{N}(0, \sigma_\theta^2)$. The IV-to-exposure effect ($\gamma_k$) and the CHP effect ($\alpha_k$) are correlated, and *i.i.d.* with a bivariate normal distribution:

$$\begin{pmatrix} \gamma_k \\ \alpha_k \end{pmatrix} \sim \mathcal{N}\left(\mathbf{0}, \begin{pmatrix} \sigma_\gamma^2, & \rho_{\alpha\gamma}\,\sigma_\gamma\sigma_{\alpha_0} \\ \rho_{\alpha\gamma}\,\sigma_\gamma\sigma_{\alpha_0}, & \sigma_{\alpha_0}^2 \end{pmatrix}\right), \tag{4}$$

where $\rho_{\alpha\gamma}$ is the correlation between $\gamma_k$ and $\alpha_k$.

**The decomposition of CHP effects.** From Eq. (4), we reparameterize $\gamma_k$ and $\alpha_k$ as follows

$$\gamma_k \sim \mathcal{N}(0, \sigma_\gamma^2), \alpha_k = \rho_{\alpha\gamma} \cdot \frac{\sigma_{\alpha0}}{\sigma_\gamma} \cdot \gamma_k + \sqrt{1 - \rho_{\alpha\gamma}^2} \cdot \sigma_{\alpha0} \cdot Z_k \overset{\text{def}}{=} \delta \cdot \gamma_k + \widetilde{\alpha}_k, \tag{5}$$

where $Z_k$ follows a standard normal distribution, $Z_k \sim \mathcal{N}(0,1)$ and $Z_k \perp\!\!\!\perp \gamma_k$, and $\delta = \rho_{\alpha\gamma} \frac{\sigma_{\alpha0}}{\sigma_\gamma}$. Equation (5) decomposes the CHP effect $\alpha_k$ into two parts, with one being proportional to $\gamma_k$ and the other part being independent of $\gamma_k$, i.e., $\widetilde{\alpha}_k \perp\!\!\!\perp \gamma_k$. The decomposition in Eq. (5) can also be viewed as a linear regression of $\alpha_k$ regressed on $\gamma_k$ with $\widetilde{\alpha}_k$ being the residuals. Let $\widetilde{\alpha}_k \sim \mathcal{N}(0, \sigma_\alpha^2)$. We call $\widetilde{\alpha}_k$ as the orthogonal projection of CHP. We can further parameterize the effect size of IV-to-outcome for IV $k$ as in Eq. (2). Therefore, identifying the IVs with CHP effects in Eq. (2) is equivalent to identifying the IVs with non-zero projected CHP, namely $\widetilde{\alpha}_k \neq 0$. The estimation of causal effect $\beta_1$ is based on IVs with $\widetilde{\alpha}_k = 0$.

We further introduce a latent indicator $\eta_k$ for each IV $k$, with $\eta_k = 1$ for IVs with non-zero CHP effects. We impose the following spike-slab prior[16,58] on $\widetilde{\alpha}_k$:

$$\widetilde{\alpha}_k \sim \begin{cases} \mathcal{N}(0, \sigma_\alpha^2), & \eta_k = 1 \\ \delta_0(\alpha_k), & \eta_k = 0, \end{cases}$$

where $\delta_0$ denotes the Dirac delta function at zero, and $\eta_k$ follows a Bernoulli distribution with $\eta_k \sim \omega^{\eta_k}(1-\omega)^{1-\eta_k}$. Then, Eq. (2) can be written as

$$\Gamma_k | \beta_1, \beta_2, \gamma_k, \eta_k, \tau_1^2, \tau_2^2 \sim \begin{cases} \mathcal{N}(\beta_1 \gamma_k, \tau_1^2), & \eta_k = 0 \\ \mathcal{N}(\beta_2 \gamma_k, \tau_2^2), & \eta_k = 1, \end{cases} \tag{6}$$

where $\tau_1^2 = \sigma_\theta^2$ for IVs with potential UHP only and $\tau_2^2 = \sigma_\theta^2 + \sigma_\alpha^2$ with both potential UHP and CHP. Following existing literature[13,14], our model also assumes that all IVs could have potential UHP while only a sparse proportion of IVs have CHP. As a consequence of the assumption, the variability of $\Gamma_k$ is larger for the $\beta_2$ group of IVs than the $\beta_1$ group because of the existence of $\widetilde{\alpha}_k$. Thus, in Eq. (6), $\tau_2^2 > \tau_1^2$. Since both $\tau_1^2$ and $\tau_2^2$ are model parameters, we can obtain their estimates using MCMC and use them to identify $\hat{\beta}_1$ (see Supplementary Materials).

To promote the computational efficiency in low-signal-noise-ratio regime, we expand the original distribution (6) as follows[59,60]:

$$\Gamma_k | \beta_1, \beta_2, \gamma_k, \eta_k, \tau_1^2, \tau_2^2, \xi^2 \sim \begin{cases} \mathcal{N}(\beta_1 \gamma_k, \xi^2 \tau_1^2), & \eta_k = 0 \\ \mathcal{N}(\beta_2 \gamma_k, \tau_2^2), & \eta_k = 1, \end{cases} \tag{7}$$

where $\xi^2$ is an expanded parameter with a non-informative prior. By combing Eqs. (3) and (7), we build the Bayesian hierarchical model with conjugate priors for hyper parameters, $\sigma_\gamma^2 \sim \mathcal{IG}(a_\gamma, b_\gamma)$, $\tau_1^2 \sim \mathcal{IG}(a_{\tau1}, b_{\tau1})$, $\tau_2^2 \sim \mathcal{IG}(a_{\tau2}, b_{\tau2})$, and $\omega \sim Beta(a, b)$.

### Accounting for LD

We expand the MR-CUE model to allow for correlated IVs by modeling their LD structure. We model the estimated effect sizes in both exposure and outcome diseases/traits with approximated multivariate normal distributions[61] as follows,

$$\begin{aligned} \widehat{\boldsymbol{\gamma}} | \boldsymbol{\gamma}, \widehat{\mathbf{R}}, \widehat{\mathbf{S}}_\gamma &\sim \mathcal{N}(\widehat{\mathbf{S}}_\gamma \widehat{\mathbf{R}} \widehat{\mathbf{S}}_\gamma^{-1} \boldsymbol{\gamma}, \widehat{\mathbf{S}}_\gamma \widehat{\mathbf{R}} \widehat{\mathbf{S}}_\gamma), \\ \widehat{\boldsymbol{\Gamma}} | \boldsymbol{\Gamma}, \widehat{\mathbf{R}}, \widehat{\mathbf{S}}_\Gamma &\sim \mathcal{N}(\widehat{\mathbf{S}}_\Gamma \widehat{\mathbf{R}} \widehat{\mathbf{S}}_\Gamma^{-1} \boldsymbol{\Gamma}, \widehat{\mathbf{S}}_\Gamma \widehat{\mathbf{R}} \widehat{\mathbf{S}}_\Gamma), \end{aligned} \tag{8}$$

where $\widehat{\boldsymbol{\gamma}} = [\widehat{\gamma}_1, \ldots, \widehat{\gamma}_p]^T$ and $\widehat{\boldsymbol{\Gamma}} = [\widehat{\Gamma}_1, \ldots, \widehat{\Gamma}_p]^T$ are vectors for the marginal effect sizes in exposure and outcome diseases/traits, respectively; $\widehat{\mathbf{S}}_\gamma = \text{diag}([\widehat{\mathbf{s}}_{\gamma_1}, \cdots, \widehat{\mathbf{s}}_{\gamma_p}])$ and $\widehat{\mathbf{S}}_\Gamma = \text{diag}([\widehat{\mathbf{s}}_{\Gamma_1}, \cdots, \widehat{\mathbf{s}}_{\Gamma_p}])$ are the corresponding diagonal matrices for standard errors; and $\widehat{\mathbf{R}} \in \mathbb{R}^{p \times p}$ is the estimated correlation matrix among all selected IVs. In the approximated distributions in Eq. (8), all quantities except for $\widehat{\mathbf{R}}$ can be obtained from summary-level GWAS results while $\widehat{\mathbf{R}}$ is estimated using an independent reference panel data.

**Estimating LD matrix from a reference panel.** To estimate the LD matrix, we used independent reference panel data from the following sources: UK10K Project (Avon Longitudinal Study of Parents and Children, ALSPAC[62], and TwinsUK[63]) merged with European-ancestry samples in 1000 Genome Project Phase 3[64]. There are 4284 individuals in total. We conducted strict quality control for the reference data using PLINK[65] and GCTA[66]. We removed the individuals with genotype missing rates greater than 5%, and further removed one pair of individuals that have genetic relatedness larger than 0.05. Since both ALSPAC and TwinsUK cohorts contain non-European samples, we further performed the principal components analysis (PCA)[67] followed by the analysis of hierarchical clustering on principal components (HCPC)[68] to extract and restrict the analysis to samples from European ancestries. After data pre-processing, roughly 3700 samples were retained as the reference panel data.

Often it is useful to define approximately independent LD blocks a priori. Here we used LDetect[69] based on an efficient signal processing approach for choosing segment boundaries between blocks. Consequently, LDetect partitioned the entire genome into 1703 and 1445 independent blocks for European and Asian populations, respectively (http://bitbucket.org/nygcresearch/ldetect-data). For each LD block, we calculated the empirical correlation matrix and further applied a simple shrinkage correlation estimator[70] to obtain

$$\widehat{\mathbf{R}}^{(l)} = \lambda \widehat{\mathbf{R}}_{\text{emp}}^{(l)} + (1 - \lambda)\mathbf{I}^{(l)}, \tag{9}$$

where $\widehat{\mathbf{R}}_{emp}^{(l)} \in \mathbb{R}^{p_l \times p_l}$ was the empirical correlation matrix for the $l$-th block in the panel data and $\lambda \geq 0$ was a shrinkage parameter. By obtaining all $\widehat{\mathbf{R}}^{(l)}$s, $l = 1, ..., L$, we could further obtain $\widehat{\mathbf{R}} = \text{diag}(\widehat{\mathbf{R}}^{(l)}) \in \mathbb{R}^{p \times p}$ with $\sum_{l=1}^{L} p_l = p$. Here we fixed the shrinkage parameter $\lambda$ at 0.85[8].

**A group spike-slab prior.** For IVs in moderate-to-strong LD, if there is a single variant $k$ with a non-zero CHP effect, the CHP effect for other nearby variants in the block would be also non-zero. In our analyses, genetic variants across the genome can be partitioned into independent blocks. IVs from different blocks could be roughly taken as independent. Thus, the projected $\widetilde{\alpha}_k$ is estimated in a group manner. We introduce a group-level latent status $\eta_l$, indicating whether IVs within the $l$-th block having non-zero CHP effects and assigning a group-level spike-slab prior as follows:

$$\widetilde{\alpha}_{lk} \sim \begin{cases} \mathcal{N}(0, \sigma_\alpha^2), & \eta_l = 1 \\ \delta_0(\alpha_{lk}), & \eta_l = 0, \end{cases} \qquad (10)$$

where $\eta_l = 1$ implies the IVs within the $l$-th block having non-zero projected CHP effects and $\eta_l = 0$ means the projected CHP effects being all zero for IVs in the block. Here, $\eta_l$ is a Bernoulli random variable with probability $\omega$ being 1, $\eta_l \sim \omega^{\eta_l}(1 - \omega)^{1 - \eta_l}$.

Considering IVs in LD, we have the following mixture distribution for $\Gamma_{lk}$ that is similar to Eq. (7):

$$\Gamma_{lk} | \beta_1, \beta_2, \gamma_{lk}, \eta_l, \tau_1^2, \tau_2^2, \xi^2 \sim \begin{cases} \mathcal{N}(\beta_1 \gamma_{lk}, \xi^2 \tau_1^2), & \text{if } \eta_l = 0 \\ \mathcal{N}(\beta_2 \gamma_{lk}, \tau_2^2), & \text{if } \eta_l = 1. \end{cases} \qquad (11)$$

## Accounting for sample overlap

When IV-to-exposure and IV-to-outcome summary statistics are taken from biobank-sized or consortia-based GWASs with potential overlapping samples, we need to account for the potential additional correlations. To allow overlapping samples in GWAS for both diseases/traits, we could rewrite the distribution for summary statistics in Eq. (8) as a joint distribution and propose the following Bayesian hierarchical model for correlated IVs with overlapping samples,

$$\begin{pmatrix} \widehat{\boldsymbol{\gamma}} \\ \widehat{\boldsymbol{\Gamma}} \end{pmatrix} \sim \mathcal{N}\left( \begin{pmatrix} \widehat{\mathbf{S}}_\gamma \widehat{\mathbf{R}} \widehat{\mathbf{S}}_\gamma^{-1} \boldsymbol{\gamma} \\ \widehat{\mathbf{S}}_\Gamma \widehat{\mathbf{R}} \widehat{\mathbf{S}}_\Gamma^{-1} \boldsymbol{\Gamma} \end{pmatrix}, \begin{pmatrix} \widehat{\mathbf{S}}_\gamma & \mathbf{0} \\ \mathbf{0} & \widehat{\mathbf{S}}_\Gamma \end{pmatrix} \left( \mathbf{R}_e \otimes \widehat{\mathbf{R}} \right) \begin{pmatrix} \widehat{\mathbf{S}}_\gamma & \mathbf{0} \\ \mathbf{0} & \widehat{\mathbf{S}}_\Gamma \end{pmatrix} \right)$$

$$\Gamma_{lk} | \beta_1, \beta_2, \gamma_{lk}, \eta_l, \tau_1^2, \tau_2^2, \xi^2 \stackrel{iid}{\sim} \left\{ \mathcal{N}(\beta_1 \gamma_{lk}, \xi^2 \tau_1^2) \right\}^{(1-\eta_l)} \left\{ \mathcal{N}(\beta_2 \gamma_{lk}, \tau_2^2) \right\}^{\eta_l}, \qquad (12)$$

$$\gamma_{lk} | \sigma_\gamma^2 \stackrel{iid}{\sim} \mathcal{N}(0, \sigma_\gamma^2), \quad \eta_l | \omega \stackrel{iid}{\sim} \omega^{\eta_l}(1-\omega)^{1-\eta_l},$$

$$\sigma_\gamma^2 \sim \mathcal{IG}(a_\gamma, b_\gamma), \quad \tau_1^2 \sim \mathcal{IG}(a_{\tau 1}, b_{\tau 1}), \quad \tau_2^2 \sim \mathcal{IG}(a_{\tau 2}, b_{\tau 2}),$$

$$\Pr(\xi^2) \propto \frac{1}{\xi^2}, \quad \omega \sim Beta(a, b),$$

where $\otimes$ denote the Kronecker product and $\mathbf{R}_e = \begin{bmatrix} 1 & \rho_e \\ \rho_e & 1 \end{bmatrix}$ is the correlation matrix that accounts for sample overlap. Here, the correlation due to sample overlap $\rho_e$ can be estimated using summary statistics among independent variants with no associations to both exposure and outcome diseases/traits.

Since the estimated LD matrix is block-diagonal, the resulting Gibbs sampler can be performed in a parallel manner for each block. The algorithmic details are given in the Supplementary Materials.

## Generation of summary statistics in the simulation studies

We generated the summary statistics using simulated individual-level data. We first simulated genotypes $\mathbf{G}_x \in \mathbb{R}^{n_x \times p}$, $\mathbf{G}_y \in \mathbb{R}^{n_y \times p}$ and $\mathbf{G}_r \in \mathbb{R}^{n_r \times p}$ for both exposure and outcome as well as for an independent reference data, respectively, where $n_x$, $n_y$, and $n_r$ were the corresponding sample sizes and $p$ was the total number of IVs. We set the

number of blocks $L$ to be 100 or 200, and the number of IVs within a block to be 10, respectively. Correspondingly, the number of IVs was either 1000 or 2000. For all simulations, we considered $n_x = 50,000$, $n_y = 50,000$ and $n_r = 4000$.

We then generated a data matrix from a multivariate normal distribution $\mathcal{N}(\mathbf{0}, \boldsymbol{\Sigma}(r))$, where $r \in \{0.4, 0.8\}$ represented the autoregressive correlation among IVs. We simulated genotype matrix by categorizing data matrices into dosage values $\{0, 1, 2\}$ according to minor allele frequency that is uniformly distributed in $[0.05, 0.5]$. We then considered the following structural model to generate individual-level data

$$\begin{aligned} \mathbf{x}_x &= \mathbf{G}_x \boldsymbol{\gamma} + \mathbf{U}_x \boldsymbol{\psi}_x + \boldsymbol{\epsilon}_{x_x}, \\ \mathbf{x}_y &= \mathbf{G}_y \boldsymbol{\gamma} + \mathbf{U}_y \boldsymbol{\psi}_x + \boldsymbol{\epsilon}_{x_y}, \\ \mathbf{y} &= \beta_1 \mathbf{x}_y + \mathbf{G}_y \boldsymbol{\alpha} + \mathbf{G}_y \boldsymbol{\theta} + \mathbf{U}_y \boldsymbol{\psi}_y + \boldsymbol{\epsilon}_y, \end{aligned} \qquad (13)$$

where $\mathbf{U}_x \in \mathbb{R}^{n_x \times q}$ and $\mathbf{U}_y \in \mathbb{R}^{n_y \times q}$ are the matrices for $q$ confounders in the samples from IV-to-exposure and IV-to-outcome, respectively, $\boldsymbol{\psi}_x \in \mathbb{R}^{q \times 1}$ and $\boldsymbol{\psi}_y \in \mathbb{R}^{q \times 1}$ are the corresponding vector of coefficients, $\mathbf{x}_x$ and $\mathbf{x}_y$ are exposure traits in two samples, $\boldsymbol{\epsilon}_{x_x} \in \mathbb{R}^{n_x \times 1}$, $\boldsymbol{\epsilon}_{x_y} \in \mathbb{R}^{n_y \times 1}$, and $\boldsymbol{\epsilon}_y \in \mathbb{R}^{n_y \times 1}$ are the random errors, and $\beta_1$ is the causal effect of interest. In all simulations, we considered $q = 50$ and each column of $\mathbf{U}_x$ and $\mathbf{U}_y$ was sampled from a standard normal distribution. The coefficients of these confounders, $\boldsymbol{\psi}_x$ and $\boldsymbol{\psi}_y$, were sampled from a bivariate normal distribution $\mathcal{N}(\mathbf{0}, \boldsymbol{\Sigma}_\psi)$, where $\boldsymbol{\Sigma}_\psi$ was a two-by-two matrix with diagonal elements of 1 and off-diagonal elements of 0.8. For CHP effects, we assumed $\gamma_k$ and $\alpha_k$ following a bivariate normal distribution $\mathcal{N}(\mathbf{0}, \boldsymbol{\Sigma}(\rho_{\alpha\gamma}))$. We considered $\alpha_k$ to be sparse, i.e., only 10% of $\alpha_k$ was sampled from the bivariate normal distribution and the others were zero. For UHP, we assumed $\theta_k$ to be dense and follow an independent normal distribution, $\mathcal{N}(0, \sigma_\theta^2)$.

We further performed the single-variant analysis to obtain summary statistics, $\{\widehat{\gamma}_k, \widehat{s}_{\gamma_k}\}$ and $\{\widehat{\Gamma}_k, \widehat{s}_{\Gamma_k}\}$, $\forall k = 1, ..., p$, for both exposure and outcome, respectively. In the simulation study, we controlled the magnitudes for $\boldsymbol{\gamma}$, $\boldsymbol{\alpha}$ and $\boldsymbol{\theta}$ using $h_\gamma^2 = \frac{var(\beta_1 \mathbf{G}_y \boldsymbol{\gamma})}{var(\mathbf{y})}$, $h_\alpha^2 = \frac{var(\mathbf{G}_y \boldsymbol{\alpha})}{var(\mathbf{y})}$ and $h_\theta^2 = \frac{var(\mathbf{G}_y \boldsymbol{\theta})}{var(\mathbf{y})}$, respectively. We considered $h_\gamma^2 = 0.1$ and varied $h_\theta^2 \in \{0.02, 0.05\}$ and $h_\alpha^2 \in \{0.05, 0.1\}$ to evaluate the performance of MR-CUE in selecting/identifying IVs with CHP effects and in the control of type I error rates. To further examine the power, we varied $h_\gamma^2$ in a sequence of values from 0 to 0.1 while fixing other parameters.

## Generation of summary statistics for reverse causation analysis

We considered the following structural model to generate individual-level data that is similar to existing work[13]:

$$\mathbf{x}_x = \mathbf{G}_x \boldsymbol{\gamma} + \boldsymbol{\epsilon}_{x_x}, \quad \mathbf{x}_y = \mathbf{G}_y \boldsymbol{\gamma} + \boldsymbol{\epsilon}_{x_y}, \quad \mathbf{y} = \beta_1 \mathbf{x}_y + \mathbf{G}_y \boldsymbol{\theta} + \boldsymbol{\epsilon}_y, \qquad (14)$$

where $\boldsymbol{\gamma}$ and $\boldsymbol{\theta}$ are from two independent normal distributions. In this simulation, we first controlled the heritability of exposure and outcome, denoted as $h_x^2$ and $h_y^2$, respectively. We further assumed that 20% of the outcome heritability, $h_y^2$, can be explained by the causal effect ($\beta_1$) of exposure on outcome. Thus, we have three quantities below

$$h_x^2 \stackrel{\text{def}}{=} \frac{var(\mathbf{G}_x \boldsymbol{\gamma})}{var(\mathbf{x}_x)}, \quad h_y^2 \stackrel{\text{def}}{=} \frac{var(\beta_1 \mathbf{G}_y \boldsymbol{\gamma} + \mathbf{G}_y \boldsymbol{\theta})}{var(\mathbf{y})} \quad \text{and} \quad \frac{var(\beta_1 \mathbf{G}_y \boldsymbol{\gamma})}{var(\mathbf{y})} = \frac{h_y^2}{5}.$$

We set $h_y^2 = 0.25$, $h_x^2 = 0.3$, and only 5% of $\boldsymbol{\gamma}$ being non-zero. We fixed $r = 0.4$, $p = 2000$, and $\rho_{\alpha\gamma} = 0.2$. To examine reverse causality, we applied MR-CUE and other methods to assess the causal effects in both directions for 100 simulated replicates. By varying significance

thresholds, we obtained the ROC curves for true positives vs. false positives averaged over the 100 replicates.

## Reporting summary

Further information on research design is available in the Nature Research Reporting Summary linked to this article.

## Data availability

The reference panel is the merged genotype data from UK10K and 1000 Genome Project Phase 3, available for download from the European Genome-Phenome Archive (https://www.ebi.ac.uk/ega/) with ID EGAD00001000776. The LD estimates using UK10K genotype data for the list of SNPs from HapMap Project Phase 3 (HapMap3) can be download at https://zenodo.org/record/7152063. All GWAS summary statistics used in this study are publicly available. GWAS summary statistics for IL-6 are available at http://www.phpc.cam.ac.uk/ceu/proteins/. GWAS summary statistics for T2D in the European population can be obtained at http://diagram-consortium.org/downloads.html. GWAS summary statistics for T2D in the East Asian population can be accessed here https://blog.nus.edu.sg/agen/summary-statistics/t2d-2020/. Other summary statistics are publicly available from the studies as referenced in Supplementary Data 1.

## Code availability

The MR-CUE method is implemented in an open-source, publicly available R package that is available at https://github.com/QingCheng0218/MR.CUE[71]. The code to reproduce the analysis can be found at https://github.com/QingCheng0218/MR.CUE/tree/main/simulation.

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

## Acknowledgements

The computational work for this article was partially performed on resources of the National Supercomputing Centre, Singapore (https://www.nscc.sg). The research of L.C. was supported by NIH 2R01GM108711, R35ES028379, and 1R01CA229618. The research of J.L. was supported by AcRF Tier 2 grant (MOET2EP20220-0009) from the Ministry of Education, Singapore, and Duke-NUS/Khoo Bridge Funding Award (Duke-NUS-KBrFA/2020/0034).

## Author contributions

L.C. and J.L. conceived the design of the study and provided funding support. Q.C. undertook all the statistical and computational analyses, developed the software tool with assistance from X.Z.; L.C. and J.L. wrote the first draft of the manuscript. Q.C., X.Z., L.C., and J.L. provided comments to refine the manuscript and approved the final version.

## Competing interests

The authors declare no competing interests.
