## [Peer Review File · Nature Communications]

Mendelian randomization accounting for complex correlated horizontal pleiotropy while elucidating shared genetic etiologyREVIEWER COMMENTS

Reviewer #1 (Remarks to the Author):

This paper proposes a new method MR-CUE to estimate the causal effect of a heritable trait using Mendelian Randomization and focuses on adjusting for correlated pleiotropy of the genetic variants. It works on an important problem and the empirical studies on IL-6 and Type-II diabetes are interesting. The paper has also conducted a thorough evaluation of MR-CUE with both simulations and negative control traits. My main concern is in the novelty and a few details of their method.

1. Novelty over existing literature. Though the problem of pleiotropy when InSIDE is violated is a challenging problem, there have already been several papers proposing new methods on it besides the two papers cited in the paper. The statistical models in some of these papers look very similar to that in MR-CUE, and some methods are more general. For instance, MRMix (Qi and Chatterjee 2019) uses Gaussian mixtures to model the SNP marginal effects which can account for both CHP and UHP SNPs. The joint Gaussian component for CHP looks similar to formula (4) in the paper. The contamination mixture (Burgess et. al. 2020) uses a contamination mixture model to deal with correlated pleiotropy. MR-Clust (Foley et. al., 2020) can identify multiple pleiotropic pathways by a clustering algorithm. GRAPPLE (Wang et. al.2021) uses a very similar model and discusses the existence of multiple confounding pathways and also discussed identifying causal direction. MR-PATH (Iong et. al.2020) proposes a similar hierarchical mixture model as this paper but allows the discovery of multiple pathways. Based on this long list of existing literature, I'm not sure about the main contribution of this paper in terms of methodology. I think the authors may need to address the new ingredients in their method and why they are important, and may also want to compare the performance of some of these new methods.

2. Multiple confounding pathways. I'm wondering how MR-CUE performs when there are multiple confounding pathways since this problem has been addressed in other papers (whether MR-CUE is robust to the violation of its model). The authors may modify their simulations to address that aspect.

3. Identification of the causal effect. When there is correlated pleiotropy, how does MR-CUE distinguish β_1 from β_2 in the model (2) and make sure that β_1 is the causal effect instead of β_2 ? Is it by assuming that the variability of γ_k is larger for the β_2 group of SNPs than the β_1 group because of the existence of $\tilde{\alpha}_k$? Or because the authors assume that the proportions of IV with no CHP is larger? I think the first assumption may not always make sense and the second assumption can be violated depending on how SNPs are selected, especially when there is more than one confounding pathway.

4. SNP selection. The authors need to provide more details on how MR-CUE selection instruments. Does MR-CUE only use strongly associated SNPs? If not, what's the cutoff? Are both SNPs kept as IV even when their LD is 0.99? The SNP selection procedure can substantially affect the performance of an MR method, and there can be Winner's curse and weak instrument bias. It can also greatly affect the computational cost. Under what circumstance will MR-CUE gain much using correlated SNPs instead of just independent SNPs as IV?

5. In the Type-II diabetes example, there seem to be many exposures that all affect Type-II diabetes and those exposures may confound with each other. How would MR-CUE result compare with a multivariable MR analysis that can simultaneously consider the effect and confounding of all exposures together?

Reviewer #2 (Remarks to the Author):

My comments are included in an attached document.

Mendelian randomization informs shared genetic etiology underlying exposure and outcome by interrogating correlated horizontal pleiotropy

Qing Cheng, Lin S. Chen, Jin Liu

Paper Review

The paper proposes a novel method for identifying and correcting for pleiotropic effects in Mendelian randomization (MR). The new method, MR-CUE, improves over the literature by explicitly modelling correlated horizontal pleiotropy, in addition to uncorrelated pleiotropy, and employs a Bayesian hierarchical model to identify and exclude variants with pleiotropic effects. MR-CUE's performance is assessed in simulations, and the method is implemented in two real-data applications, to detect causal effects of interleukin-6 on 27 outcome traits and identify determinants of Type 2 diabetes in European and Asian populations.

Here are my comments on the manuscript.

- 1) I was slightly confused about the authors' model formulation: in their Equation (1) they state that $\Gamma_k = \beta_1\gamma_k + \theta_k + \alpha_k$. Then they decompose $\alpha_k = \delta\gamma_k + \tilde{\alpha}_k$. And finally, in Equation (2), they conclude that when $\tilde{\alpha}_k = 0$, one obtains $\Gamma_k = \beta_1\gamma_k + \theta_k$. Based on their derivation, shouldn't the end-result be $\Gamma_k = \beta_1\gamma_k + \theta_k + \delta\gamma_k = \beta_2\gamma_k + \theta_k$? There is a chance I may have misunderstood the authors' calculations, but it would be nice to clarify this in the manuscript.
- 2) MR-CUE performed convincingly well in the authors' simulation study and earns its place in the MR literature as a result. However, the authors' simulated data were generated by the model assumed by their method. This has become commonplace in papers proposing new MR methods, and perhaps explains why in practice, no single pleiotropy-robust method can outperform all others in all scenarios. With that in mind, I would appreciate a more extensive discussion of the MR-CUE method's assumptions and potential shortcomings. For example, does it assume linear relationships for the confounder-exposure and confounder-outcome associations? Would it perform equally well with binary outcomes? Does it implicitly rely on some version of the plurality assumption to separate valid from invalid instruments (i.e. what would happen if a large group of pleiotropic instruments acted through the same CHP pathway and therefore exhibited similar effects)?
- 3) MR-CUE identifies variants with correlated horizontal pleiotropy, then excludes them from consideration and uses the remaining variants to estimate the causal effect of interest. This is an example of post-selection inference. How does the method account for uncertainty in the SNP selection?
- 4) The comparison of different methods in the simulation study is made in terms of power and Type I error rates. It would be useful to also report the mean bias of causal effect estimates derived from each method.

- 5) In the simulations that assess reverse causation, only 5% of the genetic instruments used are assumed to have direct effects on the exposure, while 100% of those SNPs have direct effects on the outcome (since $\theta \sim N(0, \sigma_\theta^2)$). Is this simulation setting realistic?
- 6) MR methods working with correlated SNPs usually suffer from numerical instabilities when modelling SNPs with very high genetic correlations (e.g. inverting the genetic correlation matrix becomes challenging, if not impossible). Does this apply to MR-CUE? Would the authors recommend LD-pruning selected variants at a high threshold before implementing their method?
- 7) The authors' modelling framework exhibits some similarities with other Bayesian methods for robust Mendelian randomization, including BESIDE-MR¹, JAM-MR², and Carlo Berzuini's method³. I will not request that the authors include these methods in their simulation study, since BESIDE-MR is quite similar to RAPS, JAM-MR can be quite slow to run with large numbers of correlated variants and Berzuini's method requires individual-level data. Nevertheless, it could be useful if the authors commented on any similarities and differences of their modelling framework with these existing methods.
- 8) In the Type 2 diabetes application, the authors claim that their method can account for genetic confounding. Can they elaborate on this claim? This seems to me like a slightly different issue than CHP.
- 9) Minor comments:
 - a. In Figure 1(d), the authors set the G-U effect equal to 1. Can this be done without loss of generality, and if so, why?
 - b. It may be worth pointing out that the strong correlation between Covid-19 susceptibility and severity in the authors' applied example may in fact be artificial since people with more severe Covid-19 infection are also more likely to be diagnosed with Covid-19. This could be a form of selection bias, which is known to affect studies of disease progression traits such as Covid-19 severity.
 - c. How did the authors enforce the stated heritability values in their simulations (e.g. did they do so by varying the error variances?)
 - d. It would be nice to report the F statistics corresponding to the heritability values the authors used for h_y^2 .
 - e. Equation (14) seems to suggest that in the authors' two-sample design, outcome values in the second sample were generated using risk factor values in the first sample. This would induce correlation between the two samples. It would be more realistic to generate risk factor values from the second sample, using G_y, U_y , use them to generate outcome values in the second sample and then discard them.
 - f. In Figure 4a-b, the associations seem stronger in the European than in the Asian population. Is this explained by a potentially different sample size between the two populations?
 - g. Following the definition of ξ in Equation (8), please state what prior distribution was used for it in the hierarchical model (this is only mentioned later in the paper, after Equation (13)).
 - h. In Equation (13), the distribution of η is reported twice.

¹ C. Y. Shapland et al. (2022) *Profile-likelihood Bayesian model averaging for two-sample summary data Mendelian randomization in the presence of horizontal pleiotropy*. *Statistics in Medicine*.

² A. Gkatzionis et al. (2021) *Bayesian variable selection with a pleiotropic loss function in Mendelian randomization*. *Statistics in Medicine*.

³ C. Berzuini et al. (2020) *A Bayesian approach to Mendelian randomization with multiple pleiotropic variants*. *Biostatistics*.

- i. The notation with the two covariance matrices that is used for the multivariate normal distribution of \hat{z} is slightly unorthodox. Perhaps the authors could report the distribution of \hat{z}_k instead, using more conventional notation, and then separately report the covariances across k .

Reviewer #1:

1. **Novelty over existing literature.** Though the problem of pleiotropy when InSIDE is violated is a challenging problem, there have already been several papers proposing new methods on it besides the two papers cited in the paper. The statistical models in some of these papers look very similar to that in MR-CUE, and some methods are more general. For instance, MRMix (Qi and Chatterjee 2019) uses Gaussian mixtures to model the SNP marginal effects which can account for both CHP and UHP SNPs. The joint Gaussian component for CHP looks similar to formula (4) in the paper. The contamination mixture (Burgess et. al. 2020) uses a contamination mixture model to deal with correlated pleiotropy. MR-Clust (Foley et. al., 2020) can identify multiple pleiotropic pathways by a clustering algorithm. GRAPPLE (Wang et. al., 2021) uses a very similar model and discusses the existence of multiple confounding pathways and also discussed identifying causal direction. MR-PATH (Iong et. al., 2020) proposes a similar hierarchical mixture model as this paper but allows the discovery of multiple pathways. Based on this long list of existing literature, I'm not sure about the main contribution of this paper in terms of methodology. I think the authors may need to address the new ingredients in their method and why they are important, and may also want to compare the performance of some of these new methods.

Our response #1:

This is a very good question. We thank the reviewer for raising the point, and accordingly we add many more details to illustrate the motivations and innovation of the proposed method. The reviewer is correct that there are several existing work in the literature attempting to address the challenging situation – when InSIDE assumption is violated, i.e., correlated horizontal pleiotropy (CHP) exists. Most of those methods including ours are based on the modeling of mixture distributions. It is known that mixture models can be statistically difficult to estimate, especially when the number of instrumental variables (IVs) is not large. As such, the actual performance of those methods not only depends on the model assumptions but also the implementation details (Xue et al., 2021). Below we first discuss the innovation of MR-CUE in the modeling, then show detailed comparisons with other methods based on simulations.

Innovation:

1) Modeling of IV-specific CHP effects allowing multiple confounders:

Our model allows for IV-specific CHP effects (IV-to-outcome effect via confounder), while most existing methods assume CHP effect α_k being proportional to IV-to-exposure effect, γ_k (i.e., a constant ratio $\delta = \alpha_k/\gamma_k$) for all IV k . It is commonly believed that genetically-regulated genes or pathways is a source of CHP (i.e., IVs/SNPs are associated with unmeasured confounders affecting both exposure and outcome traits). See Fig. R1a for an illustration. Let us first focus on one confounder U_1 . Denote the IV-to-confounder effect as μ_{k1} , the confounder-to-exposure effect as ψ_{x1} , and the confounder-to-outcome effect as ψ_{y1} . The CHP effect is $\alpha_k = \mu_{k1}\psi_{y1}$ and the IV-to-exposure effect is $\gamma_k = \mu_{k1}\psi_{x1}$, and the constant ratio ($\delta = \alpha_k/\gamma_k = \psi_{y1}/\psi_{x1}$) holds for uncorrelated IVs if there is only one confounder. However, when there exist multiple confounders (e.g., a genetic pathway consists of several genes) for uncorrelated IVs (Fig. R1a) or when there is a single confounder but with correlated IVs (Fig. R1b, details later), the constant ratio assumption can be violated (see Table R1). In the case of multiple confounders, different IVs may be associated with different confounder genes with different levels of strengths, and the sum of CHP effects via multiple confounders $\alpha_k = \sum_{m=1}^2 \mu_{km}\psi_{ym}$ may not be proportional to the IV-to-exposure effect $\gamma_k = \sum_{m=1}^2 \mu_{km}\psi_{xm}$ for all IVs. Thus, it is necessary to model IV-specific CHP effects

besides the average/shared CHP effects for all IVs with CHP, when considering the realistic setting with multiple unmeasured confounders.

In Table R1, we provide a numerical example and calculate the ratios (α_k/γ_k) in the presence of two confounders, and the ratios are not a constant for the IVs. When accounting for CHP due to IV-associated genes and pathways, we argue that considering both the shared and the IV-specific CHP effects would improve the inference (in an analogous way as models with fixed and random effects). In contrast, most if not all existing methods aforementioned considered only the shared/averaged CHP effects of all IVs, and implicitly assumed that all IVs affect the entire confounder sets via a deterministic mechanism. The realistic modeling of IV-specific random effects due to multiple IV-associated confounders is a major innovation of the proposed method.

2) Using correlated IVs to boost power while modeling IV-specific CHP effects due to linkage disequilibrium (LD):

To boost power and improve the identification of IVs affected by pleiotropy, several existing methods extend their frameworks and allows for correlated IVs by modeling their LD. However, the inclusion of moderately to strongly correlated IVs also brought new challenges that are under-addressed in the presence of CHP. In Figure R1b, we illustrate that even if there is only one confounder associated with the IVs, when there are multiple correlated IVs (SNPs in moderate to strong LD), the CHP effect for each IV is not proportional to the IV-to-exposure effect due to LD (see Table R1b), resulting in IV-specific CHP effects.

Therefore, jointly considering 1) multiple confounders and 2) the use of correlated IVs to boost power, we propose to decompose the CHP effect of an IV k into two components, $\alpha_k = \delta\gamma_k + \tilde{\alpha}_k$. The first component ($\delta\gamma_k$) is the IV-shared CHP effect via confounder, and is proportional to IV-to-exposure strength, γ_k . The second component, $\tilde{\alpha}_k$, describes how the IV-specific correlations with confounder(s) may affect outcome (i.e., the unique effect of the IV on the confounders), and is independent of γ_k .

Figure R1. An illustration of the presence of IV-specific CHP effect: (a) when there are two confounders; (b) when there is a single confounder but IVs are correlated.

Case	Effect	IV ₁	IV ₂	...	Shared
(a)	IV→X (γ_k)	$0.07=0.2 \times 0.2 + 0.3 \times 0.1$	0.12	...	0.19
	CHP (α_k)	$0.17=0.2 \times 0.4 + 0.3 \times 0.3$	0.26	...	0.43
	α_k/γ_k	2.43	2.17	...	2.26
(b)	IV→X (γ_k)	$0.15=0.5 \times 0.5 \times 0.2 + 0.1$	0.15	...	0.30
	CHP (α_k)	$0.10=0.5 \times 0.5 \times 0.4$	0.20	...	0.30
	α_k/γ_k	0.67	1.33	...	1.00

Table R1. Summary of effects from IVs to exposure (X) and outcome (Y) and the corresponding ratios in the two cases of Fig. R1. In (a), there are two confounders, and the ratio between the effects from IV to exposure (X) and from IV to outcome (Y) is not constant, suggesting IV-specific CHP effects due to multiple confounders. In (b), there is a single confounder but IVs are correlated, and IV-specific CHP effects are also non-zero. The correlation coefficient between IV₁ and IV₂ is set to 0.5.

3) The modeling of IV-specific CHP effect allows for the inclusions of many weak instruments:

The IV/SNP selection procedure can substantially affect the performance of an MR method, and there can be Winner’s curse and weak instrument bias. The modeling of IV-specific CHP effect in MR-CUE alleviates the potential bias in the presence of many weak instruments. In the main text, we presented data analysis of negative and positive controls, and we used different thresholds of p -values to select IVs, including 10^{-4} , 5×10^{-5} , 10^{-5} , and 5×10^{-6} . As shown in Fig. S13 and S14, MR-CUE works reasonably well with the inclusion of weak instruments.

4) Mapping the IVs with estimated CHP to genes/ pathways for studying sources of confounding:

Another feature of MR-CUE is that it further maps the IVs with estimated CHP effects (at varying significance thresholds) to cis-genes and genetic pathways, while also allowing overlapping genes/pathways of multiple confounding pathways. It should be noted that the method GRAPPLE independently proposes to sequentially identify potential risk factors as confounders by adjusting the potential risk factors in a multivariate MR analysis (Wang et al., 2021). Comparing with existing method mapping pathways sequentially, MR-CUE allows IVs to be mapped to different pathways with overlapping genes. This is a practical scientific innovation.

4) Two-layers of inference for cross-population analysis:

MR-CUE paves the way for future cross-population MR analyses. Integrative and cross-population analyses using summary statistics from different populations and ethnic groups still present many challenges due to the substantially varying LD patterns, difficulties in data harmonization, study and effect heterogeneity and others. MR-CUE is robust in cross-population analyses as it offers two layers of inference – it obtains the causal effect estimation using IVs not associated with confounders, while also maps the underlying genes and pathways for IVs affected by confounding. Both layers of inference may inform cross-population effect patterns for exposure and disease of interest. Additionally, the modeling of IV-specific CHP effects alleviates biases from weak instruments and inconsistent instruments from two-sample or even multi-ethnic summary data.

Comparison with existing methods:

In Table R2, we compare MR-CUE with aforementioned existing methods from the following perspectives: whether a method accounts for uncorrelated horizontal pleiotropy (UHP) and/or CHP; whether

a method allows for multiple confounders; whether a method allows/leverages IVs in LD; whether a method considers potential IV-specific CHP effects; whether a method allows for weak instruments; whether a method further maps estimated IVs with CHP effects to shared genetic pathways between exposure and outcome traits; estimation procedure; pros/cons. Among them, MRMix uses a four-component mixture model to identify the causal effect in the group of valid IVs without distinguishing the mechanisms (UHP/CHP) of those invalid IVs (Qi and Chatterjee, 2019). The contamination mixture MR was proposed to make causal inferences allowing some IVs to be invalid (Burgess et al., 2020). In their mixture distribution, “the values estimated by invalid instruments are normally distributed about *zero* with a large standard deviation”. This specification makes it unable to account for unbalanced CHP when the SNPs with CHP effects present a systematic bias to the true causal effect. MR-Clust was proposed to identify variants that reflect distinct causal mechanisms using a Gaussian mixture model (Foley et al., 2021). In MR-Clust, it clusters variants that present distinct means and variances. However, one may not always correctly specify which cluster corresponds to the cluster of IVs with CHP effects. GRAPPLE is another method recently proposed to account for CHP (Wang et al., 2021). Similar to CAUSE, GRAPPLE considers the effect of CHP to be proportional to the IV-to-exposure effect. More recently, MR-PATH was proposed to discover multiple biological mechanisms and capture the mechanistic heterogeneity; however, MR-PATH insufficiently accounts for CHP (Iong et al., 2020). Notably, MR-Clust, GRAPPLE, and MR-PATH models multiple Gaussian components, but they may not always correctly identify the causal cluster.

Method	UHP	CHP	Multiple confounders	LD among IVs	IV-specific CHP effect	Weak instruments	Mapping of IVs with CHP	Pros/cons
MR-CUE	✓	✓	✓	Moderate	✓	✓	✓	Allow multiple confounders, control type I error rates, have high power
CAUSE	✓	✓	✗	Weak LD	✗	✗	✗	Conservative, sensitive to nuisance parameter estimation
MRMix	✓	✓	✗	No LD	✗	✗	✗	Sensitive to parameter estimation
Contamination mixture MR	✓	possible	✗	No LD	✗	✗	✗	Zero mean for the component of invalid IVs
MR-Clust	✓	✗	✗	No LD	✗	✗	✗	Sensitive variance estimation, not always identify the causal cluster
GRAPPLE	✓	✓	✓	No LD	✗	✗	✓	Conservative, need follow-up analyses adjusting risk factors
MR-PATH	✓	✗	✗	No LD	✗	✗	✗	Variance estimation challenging, not always identify causal cluster

Table R2. Summary and comparison of some existing MR methods.

Methods comparison via simulation studies:

Last but not least, we perform additional simulations to compare with competing methods and to support our arguments. We evaluate the type I error rates and powers for MR-Clust, GRAPPLE, MR-PATH and MRMix in comparison with other methods considered in the main text. As shown in Fig. R2, MR-CUE can effectively control type I error rates while improve the statistical power comparing with other existing methods in different simulation settings with varying magnitudes and levels of sparsity of CHP effects from multiple genes/pathways. Note that except for MR-CUE and MR-LDP, we performed SNP

clumping when selecting instruments for other competing methods, and this is because other methods were proposed to use independent or weakly dependent IVs (see Table R2). Overall, in the presence of CHP from multiple confounders, MR-CUE, CAUSE, GRAPPLE and cML-MA-BIC-DP could control the type I error rates while CAUSE and GRAPPLE are conservative in many settings. Among the four methods that can control the type I error rates, MR-CUE has the highest power in all settings.

Figure R2. Comparison of MR-CUE with existing methods. (a) Type I error rates for MR-CUE and other methods under combinatorial settings for h_{θ}^2 and h_{α}^2 with $\rho_{\alpha\gamma} = 0.2$, $p = 1,000$ and the sparsity of CHP effect is set to 0.1. Here p is the total number of IVs, r represents the autoregressive correlation among IVs, $\rho_{\alpha\gamma}$ is the correlation between α and γ , and h_{γ}^2 , h_{α}^2 and h_{θ}^2 are used to control the magnitudes for γ , α and θ . (b) Powers for MR-CUE and other methods under the setting: $h_{\theta}^2 = 0.02$, $h_{\alpha}^2 = 0.05$, $p = 1,000$, $r = 0.4$, $\rho_{\alpha\gamma} = 0.2$ and the sparsity of CHP effect is set to 0.1. (c) QQ plots of $-\log_{10}(p\text{-values})$ for all methods under the null from the analyses of negative controls using 5×10^{-4} as the IV selection threshold. (d) QQ plots of $-\log_{10}(p\text{-values})$ for all methods from the analyses of positive controls using 5×10^{-4} as the IV selection threshold.

Changes we made: We rewrote the Introduction section to present and emphasize the stated innovation. We also added Fig. R1 in the main text Fig. 1 as subfigures to illustrate the potential sources of IV-specific CHP effect – a major innovation of the proposed work. We added the aforementioned existing methods to the simulation studies and method comparison whenever possible/appropriate, and illustrated the advantages in the type I error rate control and power improvement of the proposed method. We rewrote the

first paragraph of the Discussion section to summarize the main points.

2. Multiple confounding pathways. I'm wondering how MR-CUE performs when there are multiple confounding pathways since this problem has been addressed in other papers (whether MR-CUE is robust to the violation of its model). The authors may modify their simulations to address that aspect.

Our response #2:

This is a very good suggestion. The multiple confounders problem is one of the main motivations of the proposed method, but is not sufficiently explained nor illustrated in the initial submission. We rewrote the introduction to address those points. According to the reviewer's suggestion, we conducted the following simulations and compared the following methods that were proposed to account for CHP in MR inference. Those comparison methods include: MR-CUE, CAUSE, cML-MA-BIC-DP and GRAPPLE.

Specifically, we generated multiple confounding pathways, and we simulated five groups of IVs being associated with five different confounders with varying effect sizes. It should be noted that this realistic simulation setting of multiple confounders is also an illustration of our innovation 1) in Response # 1 – when there are multiple IVs associated with more than one confounders, there could be IV-specific associations to the confounders resulting in IV-specific CHP effects.

In details, we simulated

$$\begin{aligned} \gamma_{kl} &\sim \mathcal{N}(0, \sigma_\gamma^2) \\ \alpha_{kl} &= \delta \cdot \gamma_{kl} + \tilde{\alpha}_{kl} \end{aligned}$$

where $\tilde{\alpha}_{kl} \sim \mathcal{N}(0, \sigma_\alpha^2)$ and $\gamma_{kl} \perp\!\!\!\perp \tilde{\alpha}_{kl}$. The five groups of IVs are associated with those five confounders with effects, $(\sigma_{\alpha_1}^2 = 0.02, \delta_1 = 0.02)$, $(\sigma_{\alpha_2}^2 = 0.01, \delta_2 = 0.05)$, $(\sigma_{\alpha_3}^2 = 0.01, \delta_3 = 0.1)$, $(\sigma_{\alpha_4}^2 = 0.005, \delta_4 = 0.5)$, and $(\sigma_{\alpha_5}^2 = 0.005, \delta_5 = 0.8)$.

As shown in Figs. R3, MR-CUE is the only method that can sharply control the type I error rates in all settings. Note that MR-CUE, CAUSE, GRAPPLE and cML-MA-BIC-DP have a reasonable control of the type I error rates, while CAUSE and GRAPPLE are conservative in many settings. cML-MA-BIC-DP allows only IVs in weak or no LD and is less powerful than MR-CUE. MR-CUE enjoys the best power among the methods that can control the type I error rates.

Figure R3. Methods comparison in simulations with multiple confounding pathways. (a) Type I error

rates of MR-CUE and other methods under combinatorial settings for h_θ^2 and h_α^2 with $p = 1,000$. (b) Powers of MR-CUE and other methods under the setting: $h_\theta^2 = 0.1$, $h_\alpha^2 = 0.05$, $p = 1,000$ and $r = 0.4$.

Changes we made: We update the simulation section and add the simulations with multiple confounders (Fig. 2b and 2d). We also discuss those points throughout the paper.

3. Identification of the causal effect. When there is correlated pleiotropy, how does MR-CUE distinguish β_1 from β_2 in the model (2) and make sure that β_1 is the causal effect instead of β_2 ? Is it by assuming that the variability of Γ_k is larger for the β_2 group of SNPs than the β_1 group because of the existence of $\tilde{\alpha}_k$? Or because the authors assume that the proportions of IV with no CHP is larger? I think the first assumption may not always make sense and the second assumption can be violated depending on how SNPs are selected, especially when there is more than one confounding pathway.

Our response #3:

This is a good question. Following existing literature (Morrison et al., 2020; Xue et al., 2021), our model also assumes that all IVs could have potential UHP effect while only a sparse proportion of IVs have CHP effect. As a consequence of the assumption, the variability of Γ_k is larger for the β_2 group of SNPs than the β_1 group because of the existence of variation in $\tilde{\alpha}_k$.

In Equation (6) of the main text, we have

$$\Gamma_k | \beta_1, \beta_2, \gamma_k, \eta_k, \tau_1^2, \tau_2^2 \sim \begin{cases} \mathcal{N}(\beta_1 \gamma_k, \tau_1^2), & \eta_k = 0 \\ \mathcal{N}(\beta_2 \gamma_k, \tau_2^2), & \eta_k = 1 \end{cases} \quad (1)$$

where $\tau_1^2 = \sigma_\theta^2$ for IVs with potential UHP only and $\tau_2^2 = \sigma_\theta^2 + \sigma_\alpha^2$ with both potential UHP and CHP. In this formulation, $\tau_2^2 > \tau_1^2$. Since both τ_1^2 and τ_2^2 are model parameters, we can obtain their estimates using MCMC.

On the other hand, the reviewer is correct that empirically it is indeed possible to have the estimated variability in IV Set 2 (UHP+CHP) being smaller than that in IV Set 1 (UHP only).

To evaluate the validity of the condition in different settings, we conducted the following simulations. For each block of IV sets l , we reparameterize γ_{kl} and α_{kl} as follows

$$\begin{aligned} \gamma_{kl} &\sim \mathcal{N}\left(0, \sigma_\gamma^2\right) \\ \alpha_{kl} &= \delta \cdot \gamma_{kl} + \tilde{\alpha}_{kl}, \end{aligned}$$

where $\tilde{\alpha}_{kl} \sim \mathcal{N}(0, \sigma_\alpha^2)$ and $\gamma_{kl} \perp\!\!\!\perp \tilde{\alpha}_{kl}$. Thus, for IVs with no CHP effect, the causal effect from exposure to outcome is given by, β_1 . For IVs with both UHP and CHP effects, the causal effect is $\beta_2 = \delta + \beta_1$, and δ is the difference/bias in estimation due to CHP. To assess the identifiability of causal effects, we vary the value of δ and the proportion of IVs with CHP. We evaluate each simulation setting with 100 replications. We examine the proportion of sets of estimates with $\hat{\tau}_1^2 < \hat{\tau}_2^2$. We set the true causal effect to be 1. In Table R3, we show that when the sparsity levels of IVs with CHP is sparse (0.1) or moderately sparse (0.3), using the condition $\hat{\tau}_1^2 < \hat{\tau}_2^2$ to identify the causal effect $\hat{\beta}_1$ works well. However, when the proportion of IVs with CHP becomes non-sparse (0.5) AND when the difference between β_1 and β_2 (i.e., δ) is large (1 or -1), the identification has some issues and the estimated β_1 start to show biases. It should be noted that when

the proportion of IVs with CHP is non-sparse, CHP and UHP effects become statistically unidentifiable – a common challenge in the existing literature. Thus, many existing methods assume that the proportion of IVs with CHP is sparse or moderately sparse (Bowden et al., 2016; Verbanck et al., 2018; Morrison et al., 2020).

As a conclusion, when the proportion of IVs with CHP effect is sparse or moderately sparse (<0.5), the condition of $\hat{\tau}_1^2 < \hat{\tau}_2^2$ works very well in identifying the true causal effects. When the proportion is non-sparse (≥ 0.5) AND the confounding bias (δ) is large, the identification condition may lead to biased estimation.

δ	Proportion of IVs with CHP: 0.1			Proportion of IVs with CHP: 0.3			Proportion of IVs with CHP: 0.5		
	$\hat{\beta}_1$	$\hat{\beta}_2$	Prop. of replications with $\hat{\tau}_1^2 < \hat{\tau}_2^2$ (%)	$\hat{\beta}_1$	$\hat{\beta}_2$	Prop. of replications with $\hat{\tau}_1^2 < \hat{\tau}_2^2$ (%)	$\hat{\beta}_1$	$\hat{\beta}_2$	Prop. of replications with $\hat{\tau}_1^2 < \hat{\tau}_2^2$ (%)
0.0010	1.004	1.007	100	0.996	1.001	100	1.006	0.491	71
0.1000	0.997	1.111	100	1.003	1.083	99	1.020	0.513	73
1.0000	0.998	1.726	100	1.046	1.498	100	1.148	0.657	55
-1.0000	0.992	0.299	100	0.952	0.482	100	0.872	0.161	72

Table R3. An evaluation of the identifiability of causal effects in various simulation settings, using the condition $\hat{\tau}_1^2 < \hat{\tau}_2^2$. Here $\sigma_\alpha^2 = 0.1, h_\gamma^2 = 0.1, h_\alpha^2 = 0.05, h_\theta^2 = 0.05, r = 0.4, p = 1,000$. Each simulation setting has 100 replications.

Changes we made: We state and explain the assumptions in detail in the Method section (page 14). We acknowledge it as a limitation in the Discussion section (page 11). We include the results in the Supplementary Materials (Table S1).

4. SNP selection. The authors need to provide more details on how MR-CUE selection instruments. Does MR-CUE only use strongly associated SNPs? If not, what’s the cutoff? Are both SNPs kept as IV even when their LD is 0.99? The SNP selection procedure can substantially affect the performance of an MR method, and there can be Winner’s curse and weak instrument bias. It can also greatly affect the computational cost. Under what circumstance will MR-CUE gain much using correlated SNPs instead of just independent SNPs as IV?

Our response #4:

We agree with the reviewer that “The SNP selection procedure can substantially affect the performance of an MR method, and there can be Winner’s curse and weak instrument bias.”

Regarding weak instruments, MR-CUE allows weak instruments and uses genetic variants passing moderate association thresholds with exposure. In the analysis of negative and positive controls, we used different p -value thresholds to select IVs, including $10^{-4}, 5 \times 10^{-5}, 10^{-5},$ and 5×10^{-6} . As shown in Fig. S13 and S14, MR-CUE works well in the presence of many weak instruments, and this is at least partially because that MR-CUE allows for IV-specific CHP effects, which also captures some random effects of IVs due to weak instrument bias.

In terms of LD thresholds for correlated IVs, we performed LD clumping to exclude variants in high LD

($r^2 \geq 0.81$) and explicitly modeled the correlated IVs using summary-level distributions. Specifically, for each LD block, we calculated the empirical correlation matrix and further applied a simple shrinkage correlation estimator to obtain

$$\widehat{\mathbf{R}}^{(l)} = \lambda \widehat{\mathbf{R}}_{\text{emp}}^{(l)} + (1 - \lambda) \mathbf{I}^{(l)}, \quad (2)$$

where $\widehat{\mathbf{R}}_{\text{emp}}^{(l)} \in \mathbb{R}^{p_l \times p_l}$ was the empirical correlation matrix for the l -th block in the reference panel data and $\lambda \geq 0$ was a shrinkage parameter. The shrinkage estimate guarantees the matrix $\mathbf{R}^{(l)}$ to be full rank. With a liberal threshold to select IVs and by allowing for correlated IVs, there are a much larger number of instruments being used in the MR inference and the identification and estimation of causal effect accounting for UHP and CHP could be more powerful, in comparison with analysis with stringent IV selection thresholds. Moreover, as explained in our response #1, since our method allows IV-specific CHP effects, MR-CUE can control the type I error rates in the presence of CHP with correlated IVs.

Next we show the power gain of MR-CUE using correlated SNPs instead of just independent SNPs as instruments in the studies of negative and positive controls. As shown in Fig. R4(a), MR-CUE could still control the type I error rates with the use of correlated SNPs as instruments. In Fig. R4(b), we observe a substantial power gain of the proposed MR-CUE with correlated IVs and with relaxed IV selection thresholds, in comparison with using only independent SNPs as IVs.

Figure R4. Comparison of MR-CUE results with correlated and independent SNPs (labeled as MR-CUE-Indep) as instruments in the studies of (a) negative control and (b) positive control. Brackets indicate different IV selection thresholds.

To minimize potential biases due to the winner's curse, we recommend selecting the IVs first using a third independent sample (Zhao et al., 2019), if possible. The proposed method allows IV-specific random (CHP) effects, and could at least partially alleviate the bias in two-sample MR analyses.

In terms of computation, as shown in Fig. R5, the computation of MR-CUE is in the order of $\mathcal{O}(p)$ and is scalable to a large number of IVs.

Figure R5. The mean of computation time over 100 iterations for different numbers of IVs, p . We completed the analysis on a Linux platform with a 2.60 GHz intel Xeon CPU E5-2690 v3 with 30720 KB cache and 96 GB RAM.

Changes we made: We add the discussion of weak instruments, LD clumping, computation, and potential bias due to the winner's curse throughout the main text. We include the figures and results in the main text and Supplementary Materials (Fig. 2g and Fig. S15).

5. In the Type-II diabetes example, there seem to be many exposures that all affect Type-II diabetes and those exposures may confound with each other. How would MR-CUE result compare with a multivariable MR analysis that can simultaneously consider the effect and confounding of all exposures together?

Our response #5:

This is a thought-provoking question. In the type 2 diabetes (T2D) analysis, we examined the causal effects and shared pathways of different exposures on T2D by applying MR-CUE to each exposure-T2D trait pair and separately estimating the causal effect from each exposure on T2D risk. We then pulled the MR results and the estimated sets of IVs with CHP across analyses of different exposures to examine shared confoundings and mechanisms. The proposed MR-CUE is a univariable MR approach.

As far, we are not aware of any multivariate MR methods allowing for CHP. This is indeed a more challenging topic. For multivariate MR methods assuming no pleiotropy, the assumption of no pleiotropy could be restrictive. Still, the multivariable approach assesses the joint and conditional effects of multiple exposures, while the univariable method focuses more on one exposure at a time. Both types of methods could be used to assess the causal effect and inform the shared etiology between multiple exposures and an outcome of interest.

Changes we made: We discuss the connection to multivariable MR analysis in the analysis of T2D (page 10).

Responses to Reviewer #2's comments:

1. I was slightly confused about the authors' model formulation: in their Equation (1) they state that $\Gamma_k = \beta_1\gamma_k + \theta_k + \alpha_k$. Then they decompose $\alpha_k = \delta\gamma_k + \tilde{\alpha}_k$. And finally, in Equation (2), they conclude that when $\tilde{\alpha}_k = 0$, one obtains $\Gamma_k = \beta_1\gamma_k + \theta_k$. Based on their derivation, shouldn't the end-result be $\Gamma_k = \beta_1\gamma_k + \theta_k + \delta\gamma_k = \beta_2\gamma_k + \theta_k$? There is a chance I may have misunderstood the authors' calculations, but it would be nice to clarify this in the manuscript.

Our response #6:

Thanks for pointing this out. In MR-CUE, we assume all variants have two potential mechanisms to affect outcome through exposure, depicted in Fig. 1b. SNPs in IV Set 1 are not associated with unmeasured confounders ($\alpha_k = 0$) and are having potential UHP effects, θ_k . SNPs in IV Set 2 are associated with the unknown confounders. CHP effect, $\alpha_k \neq 0$, is present for IVs in Set 2 only. We apologize for the confusion and have corrected Eqn. (1) as follows:

$$\Gamma_k = \begin{cases} \beta_1\gamma_k + \theta_k, & \text{if } k \in \text{IV Set 1 with no CHP} \\ \beta_1\gamma_k + \theta_k + \alpha_k, & \text{if } k \in \text{IV Set 2 with CHP.} \end{cases} \quad (3)$$

If the k -th IV belongs to Set 2, it has CHP effect, and

$$\begin{aligned} \Gamma_k &= \beta_1\gamma_k + \theta_k + \alpha_k \\ &= \beta_1\gamma_k + \theta_k + \delta\gamma_k + \tilde{\alpha}_k \\ &= \beta_2\gamma_k + \theta_k + \tilde{\alpha}_k, \end{aligned}$$

where $\tilde{\alpha}_k$ is present only when $\alpha_k \neq 0$.

Changes we made: We made a correction of Equation (1) in the main text to clarify the confusion.

2. MR-CUE performed convincingly well in the authors' simulation study and earns its place in the MR literature as a result. However, the authors' simulated data were generated by the model assumed by their method. This has become commonplace in papers proposing new MR methods, and perhaps explains why in practice, no single pleiotropy-robust method can outperform all others in all scenarios. With that in mind, I would appreciate a more extensive discussion of the MR-CUE method's assumptions and potential shortcomings. For example, does it assume linear relationships for the confounder-exposure and confounder-outcome associations? Would it perform equally well with binary outcomes? Does it implicitly rely on some version of the plurality assumption to separate valid from invalid instruments (i.e. what would happen if a large group of pleiotropic instruments acted through the same CHP pathway and therefore exhibited similar effects)?

Our response #7:

This is a valid point. Following reviewer's suggestions, we added more discussions about model assumptions of MR-CUE and its shortcomings. MR-CUE assumes that all IVs could have potential UHP effect while only a sparse proportion of IVs have CHP effect as explained in our response #3. As a consequence, the true variability of Γ_k is larger for the group of IV with CHP effects than the group without ($\tau_2^2 > \tau_1^2$). MR-CUE uses the condition ($\hat{\tau}_2^2 > \hat{\tau}_1^2$) to identify the group of IVs not affect by CHP and estimate the true causal effect. When the proportion of IVs with CHP effect is not sparse (> 50%) and when the CHP bias

(δ) is large, the estimated $\hat{\tau}_2^2$ may not always be larger than the estimated $\hat{\tau}_1^2$ (see our response #3 and Table R3.) and the identifiability could be an issue. This is a limitation of the MR-CUE method.

For non-linear confounders, we considered the following structural model to generate individual-level data

$$\begin{aligned}\mathbf{x}_x &= \mathbf{G}_x\boldsymbol{\gamma} + \exp(\kappa\mathbf{U}_x\boldsymbol{\psi}_x) + \boldsymbol{\epsilon}_{x_x}, \\ \mathbf{x}_y &= \mathbf{G}_y\boldsymbol{\gamma} + \exp(\kappa\mathbf{U}_y\boldsymbol{\psi}_x) + \boldsymbol{\epsilon}_{x_y}, \\ \mathbf{y} &= \beta_1\mathbf{x}_y + \mathbf{G}_y\boldsymbol{\alpha} + \mathbf{G}_y\boldsymbol{\theta} + \exp(\kappa\mathbf{U}_y\boldsymbol{\psi}_y) + \boldsymbol{\epsilon}_y\end{aligned}$$

where $\mathbf{U}_x \in \mathbb{R}^{n_x \times q}$ and $\mathbf{U}_y \in \mathbb{R}^{n_y \times q}$ are the matrices for q confounders for exposure and outcome, respectively. We set $\kappa = 0.1$. As shown in Fig. R6, the conclusions on the controls of type I error rates and power comparison are similar to those with linear confounders (Figures R2 and R3). MR-CUE allows for multiple confounders, and the effects of non-linear confounders are similar to those of multiple confounders.

Figure R6. Simulation results with non-linear confounders. (a) Type I error rates for MR-CUE and other methods under combinatorial settings for h_θ^2 and h_α^2 with $\rho_{\alpha\gamma} = 0.2$ and $p = 1,000$. (b) Powers for MR-CUE and other methods under the setting: $h_\theta^2 = 0.02$, $h_\alpha^2 = 0.05$, $p = 1,000$, $r = 0.4$ and $\rho_{\alpha\gamma} = 0.2$.

For binary outcome, we generated data with binary outcome using the following logistic model:

$$\begin{aligned}\mathbf{x}_x &= \mathbf{G}_x\boldsymbol{\gamma} + \mathbf{U}_x\boldsymbol{\psi}_x + \boldsymbol{\epsilon}_{x_x}, \\ \mathbf{x}_y &= \mathbf{G}_y\boldsymbol{\gamma} + \mathbf{U}_y\boldsymbol{\psi}_x + \boldsymbol{\epsilon}_{x_y}, \\ \mathbf{y} &= \text{Bernoulli}(H(\log(1/9) + \beta_1\mathbf{x}_y + \mathbf{G}_y\boldsymbol{\alpha} + \mathbf{G}_y\boldsymbol{\theta} + \mathbf{U}_y\boldsymbol{\psi}_y))\end{aligned}$$

where $H(t) = 1/(1 + \exp(t))$. The population prevalence was set to be 0.1. We first generated a large population pool of outcomes and sampled 25,000 cases and 25,000 controls for the following analysis. As shown in Fig. R7, the conclusions on the controls of type I error rates and power comparison are similar to those with continuous outcome (Figures R2 and R3).

Figure R7. Simulation results with binary outcomes. (a) Type I error rates for MR-CUE and other methods under combinatorial settings for h_0^2 and h_α^2 with $\rho_{\alpha\gamma} = 0.2$ and $p = 1,000$. (b) Powers for MR-CUE and other methods under the setting: $h_0^2 = 0.02$, $h_\alpha^2 = 0.05$, $p = 1,000$, $r = 0.4$ and $\rho_{\alpha\gamma} = 0.2$.

Lastly, we conducted additional simulations to compare with other methods when the proportion of IVs with CHP effects increases. As shown in Fig. R8, when the proportion of IVs with CHP is 0.5, the method CAUSE is conservative and is under-powered. MR-CUE, GRAPPLE and cML-MA-BIC-DP all start to show some loss in the control of type I error rates (with GRAPPLE being the worst and MR-CUE the best). MR-CUE is still the most powerful one among the methods that have a reasonable control of type I error rates.

Figure R8. Simulation results for different proportions of IVs with CHP effects. (a) Type I error rate comparison for MR-CUE and other methods under combinatorial settings for h_0^2 and h_α^2 with $\rho_{\alpha\gamma} = 0.2$, $r = 0.4$ and $p = 1,000$. (b) Power comparison for MR-CUE and other methods under the setting: $h_0^2 = 0.02$, $h_\alpha^2 = 0.05$, $p = 1,000$, $r = 0.4$, $\rho_{\alpha\gamma} = 0.2$ and with the proportion of IVs with CHP effects being 0.4.

Changes we made: We add the discussion about model assumptions in page 14 and limitations/caveats in page 11. We add the simulations of non-linear confounding effects, binary outcome, and the impact of different proportions in IVs with CHP effects in Supplementary Fig. S9-S11.

3. MR-CUE identifies variants with correlated horizontal pleiotropy, then excludes them from consideration and uses the remaining variants to estimate the causal effect of interest. This is an example of

post-selection inference. How does the method account for uncertainty in the SNP selection?

Our response #8:

We apologize for the confusion. MR-CUE is a unified method that iteratively estimates and simultaneously obtains the posterior probabilities of IVs without CHP effect and the weighted estimate of the causal effect of interest, weighing by the posterior probabilities of IVs without CHP. The estimated causal effect is a weighted estimator from all IVs (though some IVs may have very low posterior probabilities), and it is not a post-selection inference. The method estimates the causal effect accounting for estimation uncertainty in SNP class (Set 1 with UHP only versus Set 2 with UHP+CHP) by calculating and weighing the posterior probabilities.

Changes we made: We rephrase the Introduction, Methods, and Discussion section to avoid the confusions.

4. The comparison of different methods in the simulation study is made in terms of power and Type I error rates. It would be useful to also report the mean bias of causal effect estimates derived from each method.

Our response #9:

This is a valid point. To compare the biases of causal effect estimates, we show the boxplots of point estimates for competing methods in Fig. R9, with the true causal effect being $\beta_1 = 1$.

Following the reviewer’s suggestion, we further summarize the mean bias of causal effect estimates from MR-CUE and other methods in Table R4. The mean bias of MR-CUE is the smallest among all methods.

Figure R9. The boxplots of point estimates over 100 replications for competing methods, with $h_\gamma^2 = 0.1, r = 0.4, \rho_{\alpha\gamma} = 0.2, p = 2,000$.

Method	$h_{\theta}^2 = 0.02, h_{\alpha}^2 = 0.05$	$h_{\theta}^2 = 0.05, h_{\alpha}^2 = 0.1$
MR-CUE	0.071	-0.149
GRAPPLE	-0.207	-0.780
MRMix	1.940	5.470
cML-MA-BIC-DP	2.667	7.035
RAPS	4.446	6.243
MR-Clust	-5.251	-7.092
MR-LDP	-9.620	-30.535
IVW	-11.301	-8.271
MR-Egger	-11.708	-9.026
CAUSE	-24.063	-22.107

Table R4. Mean biases of causal effect estimates over 100 replications for competing methods, with $h_{\gamma}^2 = 0.1, r = 0.4, \rho_{\alpha\gamma} = 0.2, p = 2,000$, all in %.

Changes we made: We include the results in the Supplementary Materials (Table S2 and Fig. S8).

5. In the simulations that assess reverse causation, only 5% of the genetic instruments used are assumed to have direct effects on the exposure, while 100% of those SNPs have direct effects on the outcome (since $\theta \sim N(0, \sigma_{\theta}^2)$). Is this simulation setting realistic?

Our response #10:

This is a valid concern. Following existing literature (Morrison et al., 2020; Xue et al., 2021), our model assumes that all IVs could have potential uncorrelated pleiotropic effect θ while only a sparse proportion of IVs have correlated pleiotropic effect. The assumption makes UHP and CHP effects statistically identifiable.

To address the reviewer’s concern, we conducted additional simulations with varying levels of UHP sparsity and still 5% of the IVs have CHP effects. In Fig. R10, we presented methods comparison at different levels of UHP sparsity. The conclusions are unchanged.

(a) Sparse rate of θ : 0.1

(b) Sparse rate of θ : 0.3

Figure R10. The result of reverse causation with different sparse rate of θ .

Changes we made: We include the results in the Supplementary Materials (Fig. S12).

6. MR methods working with correlated SNPs usually suffer from numerical instabilities when modelling SNPs with very high genetic correlations (e.g. inverting the genetic correlation matrix becomes challenging, if not impossible). Does this apply to MR-CUE? Would the authors recommend LD-pruning selected variants at a high threshold before implementing their method?

Our response #11:

We agree that working with highly correlated SNPs may encounter the invertability problem of the genetic correlation matrix in the estimation. In our analysis, we first partitioned the entire genome into 1,703 and 1,445 independent blocks for European and Asian populations, respectively, (Berisa and Pickrell, 2016) and then applied a simple shrinkage correlation estimator (Schafer and Strimmer, 2005) as follows

$$\hat{\mathbf{R}}^{(l)} = \lambda \hat{\mathbf{R}}_{\text{emp}}^{(l)} + (1 - \lambda) \mathbf{I}^{(l)} \quad (4)$$

where $\hat{\mathbf{R}}_{\text{emp}}^{(l)} \in \mathbb{R}^{p_l \times p_l}$ is the empirical correlation matrix for the l -th block in the panel data and $\lambda \geq 0$ was a shrinkage parameter. For dense sets of SNPs, we recommend to perform LD-pruning to remove SNPs in strong LD. MR-CUE uses moderated correlated SNPs as instruments, and the power gain and the control of type I error rates are illustrated in our response #4 (Figure R4).

7. The authors' modelling framework exhibits some similarities with other Bayesian methods for robust Mendelian randomization, including BESIDE-MR¹, JAM-MR², and Carlo Berzuini's method³. I will not request that the authors include these methods in their simulation study, since BESIDE-MR is quite similar to RAPS, JAM-MR can be quite slow to run with large numbers of correlated variants and Berzuini's method requires individual-level data. Nevertheless, it could be useful if the authors commented on any similarities and differences of their modelling framework with these existing methods.

Our response #12:

Thanks for providing the reference. The method BESIDE-MR (Shapland et al., 2022) does not distinguish SNPs with CHP from those with no CHP. BESIDE-MR uses a profile likelihood method together with Bayesian model averaging to estimate the parameters. In both JAM-MR (Gkatzionis et al., 2021) and Berzuini's method (Berzuini et al., 2020), the methods focus on IVs with UHP but with no consideration of CHP. A focus and innovation of MR-CUE is the handling of IVs with CHP effects, from multiple pathways and allowing for correlated IVs.

Changes we made: We cited the above reference and discussed them as alternative methods accounting for pleiotropy. See page 5.

8. In the Type 2 diabetes application, the authors claim that their method can account for genetic confounding. Can they elaborate on this claim? This seems to me like a slightly different issue than CHP.

Our response #13:

Thanks for pointing this out. We meant to state that our method accounts for CHP effects due to IV-associated confounders. To avoid confusion, we rephrase the claim and replace "genetic confounding" with "IV-associated confounders".

Changes we made: We rephrase the statement to avoid the confusion (see page 9-10). We also check the entire manuscript and replace the word “genetic confounding” with “IV-associated confounder”.

9. Minor comments:

a). In Figure 1 (d), the authors set the G-U effect equal to 1. Can this be done without loss of generality, and if so, why?

Our response #14:

For variant k , we can parameterize IV-to-confounder effect as γ_{uk} , confounder-to-exposure effect as γ_{ek} , confounder-to-outcome effect as α_{ok} , where $\gamma_k = \gamma_{uk}\gamma_{ek}$. Since we are not interested in the point estimate of α_{ok} itself, without loss of generality, we can always reparameterize γ_{uk} to be 1, then we have $\gamma_k = \gamma_{ek}$ and $\alpha_k = \alpha_{ok}/\gamma_{uk}$. This follows from the work CAUSE (Morrison et al., 2020).

Changes we made: We rephrase the description of Figure 1 and explain the notation in the Introduction (page 3). We also cited the reference of the reparametrization (See page 4).

b)It may be worth pointing out that the strong correlation between Covid-19 susceptibility and severity in the authors’ applied example may in fact be artificial since people with more severe Covid-19 infection are also more likely to be diagnosed with Covid-19. This could be a form of selection bias, which is known to affect studies of disease progression traits such as Covid-19 severity.

Our response #15:

We thank the reviewer for pointing this out. Following the reviewer’s suggestion, we add a statement to discuss those possibilities. See page 8.

c) How did the authors enforce the stated heritability values in their simulations (e.g. did they do so by varying the error variances?)

Our response #16:

In our simulation studies, we constructed three equations related to the signal magnitude according to the definition of h_γ^2 , h_α^2 and h_θ^2 . Once we have the pre-specified values of h_γ^2 , h_α^2 and h_θ^2 , we could enforce the stated values through varying the error variances.

d) It would be nice to report the F statistics corresponding to the heritability values the authors used for h_γ^2

Our response #17:

In our simulations, h_γ^2 depicts the heritability of outcome via exposure. To quantify the strength of IV, we estimate the F statistics of the associations between IVs and exposure under different combinations of h_γ^2 and β_1 values (Pierce et al., 2011; Burgess et al., 2011). The results can be found in Table R5 with power comparison shown in Fig. R11.

$\beta_1 = 0.2$	h_γ^2	0.002	0.003	0.004	0.005	0.006	0.007	0.008	0.009	0.010
	F statistics	3.003	4.513	6.023	7.545	9.058	10.573	12.102	13.650	15.183
$\beta_1 = 0.5$	h_γ^2	0.010	0.015	0.020	0.025	0.030	0.035	0.040	0.045	0.050
	F statistics	2.917	4.397	5.904	7.425	8.966	10.527	12.082	13.684	15.313

Table R5. The mean F statistics over 500 replications under different combinations of h_γ^2 and β_1 with $h_\alpha^2 = 0.05$, $h_\theta^2 = 0.1$, $r = 0.4$, $\rho_{\alpha\gamma} = 0.2$, $p = 1,000$.

Figure R11. Power comparison for MR-CUE and other methods for different h_γ^2 and β_1 values under the setting: $h_\alpha^2 = 0.05$, $h_\theta^2 = 0.1$, $r = 0.4$, $\rho_{\alpha\gamma} = 0.2$, $p = 1,000$.

e) Equation (14) seems to suggest that in the authors' two-sample design, outcome values in the second sample were generated using risk factor values in the first sample. This would induce correlation between the two samples. It would be more realistic to generate risk factor values from the second sample, using G_y, U_y , use them to generate outcome values in the second sample and then discard them.

Our response #18:

Sorry for the confusion. We simulated exposure and outcome traits using independent genotype matrices. To clarify this, we added an additional line corresponding to exposure trait in the outcome sample in the main text as follows:

$$\begin{aligned} \mathbf{x}_x &= \mathbf{G}_x \boldsymbol{\gamma} + \mathbf{U}_x \boldsymbol{\psi}_x + \boldsymbol{\epsilon}_{x_x}, \\ \mathbf{x}_y &= \mathbf{G}_y \boldsymbol{\gamma} + \mathbf{U}_y \boldsymbol{\psi}_x + \boldsymbol{\epsilon}_{x_y}, \\ \mathbf{y} &= \beta_1 \mathbf{x}_y + \mathbf{G}_y \boldsymbol{\alpha} + \mathbf{G}_y \boldsymbol{\theta} + \mathbf{U}_y \boldsymbol{\psi}_y + \boldsymbol{\epsilon}_y, \end{aligned}$$

Changes we made: we included additional clarification. See page 16.

f) In Figure 4a-b, the associations seem stronger in the European than in the Asian population. Is this explained by a potentially different sample size between the two populations?

Our response #19:

Yes, as shown in Supplementary Tables S9 and S10, the sample sizes of exposure traits are much smaller in studies with East Asia ancestry than those in studies with European ancestry.

g) Following the definition of ζ in Equation (8), please state what prior distribution was used for it in the hierarchical model (this is only mentioned later in the paper, after Equation (13)).

Our response #20:

We added the prior after Eqn. (7) for ζ^2 in the main text. See page 14.

h) In Equation (13), the distribution of η is reported twice.

Our response #21:

Sorry. We removed the replicate of η_l in the revised manuscript.

i) The notation with the two covariance matrices that is used for the multivariate normal distribution of \hat{z} is slightly unorthodox. Perhaps the authors could report the distribution of \hat{z}_k instead, using more conventional notation, and then separately report the covariances across k .

Our response #22:

Thanks for pointing this out. This is a valid concern. The reason that we used matrix normal distribution for combined z-scores is that they have correlations due to LD and/or sample overlap. The first correlation matrix $\hat{\mathbf{R}}$ depicts the correlation due to LD among correlated variants while the second $\hat{\mathbf{R}}_e$ characterizes the correlation due to potential sample overlap. Reporting this using \hat{z}_k may lose the information about LD. To avoid the unorthodox notation, we rewrite equation (12) in the main text.

Changes we made: We rewrite the equation related to \hat{z} as follows

$$\begin{pmatrix} \hat{\boldsymbol{\gamma}} \\ \hat{\boldsymbol{\Gamma}} \end{pmatrix} \sim \mathcal{N} \left(\begin{pmatrix} \hat{\mathbf{S}}_\gamma \hat{\mathbf{R}} \hat{\mathbf{S}}_\gamma^{-1} \boldsymbol{\gamma} \\ \hat{\mathbf{S}}_\Gamma \hat{\mathbf{R}} \hat{\mathbf{S}}_\Gamma^{-1} \boldsymbol{\Gamma} \end{pmatrix}, \begin{pmatrix} \hat{\mathbf{S}}_\gamma & \mathbf{0} \\ \mathbf{0} & \hat{\mathbf{S}}_\Gamma \end{pmatrix} (\mathbf{R}_e \otimes \hat{\mathbf{R}}) \begin{pmatrix} \hat{\mathbf{S}}_\gamma & \mathbf{0} \\ \mathbf{0} & \hat{\mathbf{S}}_\Gamma \end{pmatrix} \right) \quad (5)$$

See page 15 and Eqn. (12).

Reference

- C. Berzuini, H. Guo, S. Burgess, and L. Bernardinelli. A bayesian approach to mendelian randomization with multiple pleiotropic variants. *Biostatistics*, 21(1):86–101, 2020.
- J. Bowden, G. Davey Smith, P. C. Haycock, and S. Burgess. Consistent estimation in mendelian randomization with some invalid instruments using a weighted median estimator. *Genetic epidemiology*, 40(4):304–314, 2016.
- S. Burgess, S. G. Thompson, and C. C. genetics collaboration. Avoiding bias from weak instruments in mendelian randomization studies. *International journal of epidemiology*, 40(3):755–764, 2011.
- S. Burgess, C. N. Foley, E. Allara, J. R. Staley, and J. M. Howson. A robust and efficient method for mendelian randomization with hundreds of genetic variants. *Nature communications*, 11(1):1–11, 2020.
- C. N. Foley, A. M. Mason, P. D. Kirk, and S. Burgess. Mr-clust: clustering of genetic variants in mendelian randomization with similar causal estimates. *Bioinformatics*, 37(4):531–541, 2021.
- A. Gkatzionis, S. Burgess, D. V. Conti, and P. J. Newcombe. Bayesian variable selection with a pleiotropic loss function in mendelian randomization. *Statistics in Medicine*, 40(23):5025–5045, 2021.
- D. Iong, Q. Zhao, and Y. Chen. A latent mixture model for heterogeneous causal mechanisms in mendelian randomization. *arXiv preprint arXiv:2007.06476*, 2020.
- J. Morrison, N. Knoblauch, J. H. Marcus, M. Stephens, and X. He. Mendelian randomization accounting for correlated and uncorrelated pleiotropic effects using genome-wide summary statistics. *Nature genetics*, 52(7):740–747, 2020.
- B. L. Pierce, H. Ahsan, and T. J. VanderWeele. Power and instrument strength requirements for mendelian randomization studies using multiple genetic variants. *International journal of epidemiology*, 40(3):740–752, 2011.
- G. Qi and N. Chatterjee. Mendelian randomization analysis using mixture models for robust and efficient estimation of causal effects. *Nature communications*, 10(1):1–10, 2019.
- C. Y. Shapland, Q. Zhao, and J. Bowden. Profile-likelihood bayesian model averaging for two-sample summary data mendelian randomization in the presence of horizontal pleiotropy. *Statistics in Medicine*, 41(6):1100–1119, 2022.
- M. Verbanck, C.-Y. Chen, B. Neale, and R. Do. Detection of widespread horizontal pleiotropy in causal relationships inferred from mendelian randomization between complex traits and diseases. *Nature genetics*, 50(5):693–698, 2018.
- J. Wang, Q. Zhao, J. Bowden, G. Hemani, G. Davey Smith, D. S. Small, and N. R. Zhang. Causal inference for heritable phenotypic risk factors using heterogeneous genetic instruments. *PLoS Genetics*, 17(6):e1009575, 2021.
- H. Xue, X. Shen, and W. Pan. Constrained maximum likelihood-based mendelian randomization robust to both correlated and uncorrelated pleiotropic effects. *The American Journal of Human Genetics*, 108(7):1251–1269, 2021.
- Q. Zhao, Y. Chen, J. Wang, and D. S. Small. Powerful three-sample genome-wide design and robust statistical inference in summary-data mendelian randomization. *International journal of epidemiology*, 48(5):1478–1492, 2019.

REVIEWERS' COMMENTS

Reviewer #1 (Remarks to the Author):

The authors have addressed all my previous concerns. I just have one clarification question regarding the authors' response on how MR-CUE deal with multiple confounders. In the simulations of Figure 2, the authors generate different sets of IVs corresponding to different delta values, but in the model (5) of MR-CUE seems that there is only one shared delta that is allowed. Is that correct? So MR-CUE did not handle multiple confounders explicitly, but is robust to multiple confounders because of the extra pleiotropic effect $\tilde{\alpha}_k$?

Reviewer #2 (Remarks to the Author):

I thank the authors for addressing my previous comments. I am happy with the revised version of the manuscript and would like to recommend that it is accepted for publication.

Reviewer #1:

1. The authors have addressed all my previous concerns. I just have one clarification question regarding the authors' response on how MR-CUE deal with multiple confounders. In the simulations of Figure 2, the authors generate different sets of IVs corresponding to different δ values, but in the model (5) of MR-CUE seems that there is only one shared delta that is allowed. Is that correct? So MR-CUE did not handle multiple confounders explicitly, but is robust to multiple confounders because of the extra pleiotropic effect $\tilde{\alpha}_k$?

Our response #1:

Thanks for pointing out this. In the modeling of MR-CUE, the CHP effect of each IV k is modeled as α_k . We further decompose α_k as $\alpha_k = \delta \cdot \gamma_k + \tilde{\alpha}_k$, where γ_k is the IV-to-exposure effect of each IV k , δ captures the shared or the averaged CHP effects via all confounders, and $\tilde{\alpha}_k$ describes how IV-specific correlations with confounder(s) may affect outcome. See Page 12 of main text, Equation (5).

Reviewer is correct that we did not explicitly model the effect of each confounder. Instead, we model the effect of each IV on outcome via IV-specific correlation to the confounder set. In the MR-CUE analysis, we are interested in 1) estimating the causal effect β_1 adjusting/accounting for confounders, and 2) identifying/estimating which IV is associated with confounders. For achieving either goal, we argue that explicitly modeling the effect of each confounder is not needed, and each confounder's effect may not be estimable/identifiable if there are many confounders. Our current modeling is sufficient to allow multiple confounders having different effects on outcome.

An analogy is that in estimating linear regression models, if there are multiple unmeasured confounders, one often does not need to explicitly model each confounder's effect to adjust for confounding. Instead, one could model the random-effect of each observation due to unmeasured confounder(s). In MR-CUE, we model the IV-specific correlation to the confounder(s) as a random effect that allows each IV to perturb the confounder set differently. The reviewer is correct that MR-CUE "is robust to multiple confounders because of the extra pleiotropic effect $\tilde{\alpha}_k$ ".

In the simulations of Figure 2, we generated different δ values for different IVs. In Figure R1(a) and Table R1(a) of the previous response letter, we illustrated that in the presence of multiple confounders, the common δ value can be taken as the shared/averaged CHP effects of all IVs. When $\delta \neq 0$, the overall confounding effects on outcome is non-zero. The $\tilde{\alpha}_k$ part models the IV-specific deviation from the the shared/averaged δ . In summary, MR-CUE models IV-specific correlations to the confounder set and is robust to the presence of multiple confounders without needing to explicitly model each confounder.

Changes we made: We add the above discussion and clarification on page 4, line 122-125.